# Self-trapped state enabled filterless narrowband photodetections in 2D layered perovskite single crystals

Junze Li[1], Jun Wang[1], Jiaqi Ma[1], Hongzhi Shen[1], Lu Li[1], Xiangfeng Duan [2] & Dehui Li [1,3]

Filterless narrowband photodetectors can realize color discrimination without filter or bulk spectrometer, thus greatly reducing the system volume and cost for many imaging applications. Charge collection narrowing has been demonstrated to be a successful approach to achieve filterless narrowband photodetections; nevertheless, it sacrifices the sensitivity of the photodetectors. Here we show a highly tunable narrowband photodetector based on two-dimensional perovskite single crystals with high external quantum efficiency (200%), ultralow dark current ($10^{-12}$ A), and high on–off ratio ($10^3$). The spectral response of the narrowband photodetectors can be continuously tuned from red to blue with all full-width at half-maximum < 60 nm and especially < 20 nm in blue wavelength range. The excellent performance can be ascribed to self-trapped states within bandgap and extremely low electrical conductivity in the out-of-plane direction. Our findings open the exciting potential of 2D perovskites for next-generation optoelectronics.

[1] School of Optical and Electronic Information, Huazhong University of Science and Technology, Wuhan 430074, China. [2] Department of Chemistry and Biochemistry, University of California, Los Angeles, California 90095, USA. [3] Wuhan National Laboratory for Optoelectronics, Huazhong University of Science and Technology, Wuhan 430074, China. Correspondence and requests for materials should be addressed to D.L. (email: dehuili@hust.edu.cn)

Photodetectors are indispensable components for imaging, communication, and biomedical sensing systems[1-3], and can be classified as broadband[4,5] and narrowband[6,7] in terms of the width of their spectral response range. Although broadband photodetectors are capable of detecting a broad spectral range of light from infrared[8], visible[9] to ultraviolet light[10], and even to X-ray[11], the narrowband ones can only sense a small spectral range of light and thus realize the color discrimination[12,13]. The ability of spectral discrimination for narrowband photodetectors makes them attractive in many applications including biomedical sensing, machine vision, and imaging[14]. In general, four strategies have been adopted to realize narrowband photodetections, including the following: (1) combining an optical filter with a broadband photodetector[15]; (2) using an absorber with a narrowband absorption[16-18]; (3) intentionally enhancing the absorption in a selected wavelength range by the plasmonic effect[19]; and (4) manipulating the external quantum efficiency (EQE) via charge collection narrowing (CCN)[12,13]. Despite narrowband photodetectors based on these approaches have been demonstrated with an appropriate set of performance metrics, they still have some disadvantages[13]. To this end, a low-cost narrowband photodetector with a great architectural simplicity and a wider tunable spectral response range is much desired.

Recently, two-dimensional (2D) Ruddlesden–Popper perovskites have been extensively studied due to the naturally formed multiple quantum-well structure, layered characteristic, flexibly tunable bandgap[20], and the improved long-term stability compared with their three-dimensional (3D) counterparts[21,22]. The 2D layered perovskites have a general chemical formula of $R_2A_{n-1}M_nX_{3n+1}$, where R is long chain spacer cation, A is an organic cation, whereas X is a halide anion, and M is a divalent metal. Their structure can be regarded as $n$ (integer) layers of $[PbX_6]^{4-}$ octahedral sheets sandwiched by two layers of R spacer cations[23]. The organic chain works as a barrier layer, resulting in a natural multiple quantum-well structure[24]. Equipped with such a quantum-well structure, 2D perovskites exhibit a strong quantum confinement effect, an extremely large anisotropy between the in-plane and out-of-plane electrical conductivity[25], and an improved environmental stability compared with 3D perovskites due to the hydrophobicity of the organic chain R[21]. In addition, by tuning the halide compositions and layer number $n$, the energy bandgap of 2D perovskites can be continuously tuned within entire visible wavelength range[26].

In particular, the strong electron–phonon coupling leads to the formation of self-trapped states within bandgap in 2D layered perovskites[27]. It has been predicted that the self-trapping critically depends on the dimensionality of the systems. The 2D systems have a rather low or non-existent potential barrier for the self-trapping, leading to the easier formation of self-trapped states[28]. Previous studies have demonstrated that the density of self-trapped states greatly increases when the dimensionality is reduced from 3D to 2D[29,30]. This enhanced self-trapping in 2D perovskites can alter the electrical and optical properties of the 2D perovskites including the thermal activated broad white light emission[31,32] and hopping transport of carriers[33]. Importantly, those self-trapped states are excitonic in nature and possess weak optical transition strength with a longer lifetime[32]. With these unique features, self-trapped states in 2D perovskites with distinguished weak absorption peak below free exciton absorption naturally form another absorption onset to realize the CCN via recombination losses for narrowband photodetection[29], thus bypassing the necessity to specially design the active materials. Along with the extremely large anisotropy between the in-plane and out-of-plane electrical conductivity in 2D perovskites[25], we anticipate that the narrowband photodetectors of 2D perovskite single-crystal vertical structures can exhibit excellent performance. Here we report narrowband 2D perovskite single-crystal photodetectors with high EQE of 200%, ultralow dark current of $10^{-12}$ A, high on–off ratio of $10^3$, and all full-width at half-maximum (FWHM) < 60 nm across the entire visible spectrum.

## Results

**Characterizations of 2D perovskite crystals.** Figure 1a displays the photographs of the 2D $(BA)_2(MA)_{n-1}Pb_nBr_{3n+1}$ and $(BA)_2(MA)_{n-1}Pb_nI_{3n+1}$ single crystals synthesized by the solution method previously reported[34,35], where $BA = C_4H_9NH_3$ and $MA = CH_3NH_3$. By increasing the layer number $n$ and changing the chemical compositions, millimeter-size single crystals were successfully synthesized with the color gradually changing from white to orange and finally to black, reflecting the decrease of the bandgap with the increase of the layer number $n$. The corresponding absorption spectra of the as-grown single crystals shows a gradually redshift of the absorption edge with the increase of layer number $n$, agreeing well with their color change (Fig. 1b). The photoluminescence (PL) spectra of those single crystals show that the emission peak coincides with the excitonic emission peaks of 2D perovskites consistent with previous reports[26,36] and the redshift of emission peak follows the same trend as the absorption edge shift with the increasing layer number $n$ (Fig. 1b), which can be attributed to the shrinkage of the bandgap due to the reduced quantum confinement effect[37] and the reduction of exciton binding energy resulting from the reduced dielectric confinement by organic spacers[38] for a larger $n$ number. Compared with the absorption spectra, PL peaks show a blueshift for all samples, which can be ascribed to the very thick bulk crystals we used to acquire the absorption spectra. In the very thick crystals, the absorption has already saturated before the free exciton absorption peak appears, supported by the flat absorption line at the short-wavelength regime. For microplates with thickness around 100 nm or thinner, we can observe the free exciton absorption peak and the PL peak shows slightly redshift compared with the exciton absorption peak (Supplementary Fig. 1).

Powder X-ray diffraction (XRD) measurement has been further carried out to investigate the crystalline quality of the as-synthesized 2D perovskite plates (Fig. 1c). The low-angle part is magnified to identify the layer number $n$. All diffraction peaks can be indexed to the corresponding 2D $(BA)_2(MA)_{n-1}Pb_nBr_{3n+1}$ and $(BA)_2(MA)_{n-1}Pb_nI_{3n+1}$ series[35,39,40], confirming the excellent crystalline quality of the as-synthesized single crystals. For $(BA)_2(MA)Pb_2Br_7$ perovskite, it contains the impurity phase of $(BA)_2PbBr_4$; nevertheless, it would not introduce states within bandgap and thus would not affect the FWHM of the narrow spectral response. For $(BA)_2(MA)_{n-1}Pb_nI_{3n+1}$ ($n > 2$) perovskites, they indeed contain impurity phases, which would affect FWHMs of the narrowband photodetectors.

**2D perovskite-based narrowband photodetectors.** The millimeter size of the as-grown 2D perovskite plates allows us to fabricate electronic devices by placing gold foil as electrodes to investigate their optoelectronic properties. Fig. 2a displays the schematic illustration of the 2D perovskite-based vertical devices with indium tin oxide (ITO) as the bottom contact and Au as the top electrode (denoted as Config. 1). The perovskite plate was exfoliated by Scotch tape to make a fresh surface and transferred to a pre-cleaned ITO glass substrate. A gold foil was then placed on the top of the crystal as a top electrode, similar to the alignment transfer of Au electrodes in the two-probe $MoS_2$ devices[41].

Figure 2b shows the current–voltage ($I$–$V$) curves of a $(BA)_2(MA)Pb_2I_7$ plate vertical device with the plate thickness of 80 μm in dark, and under a 620 nm and a 550 nm

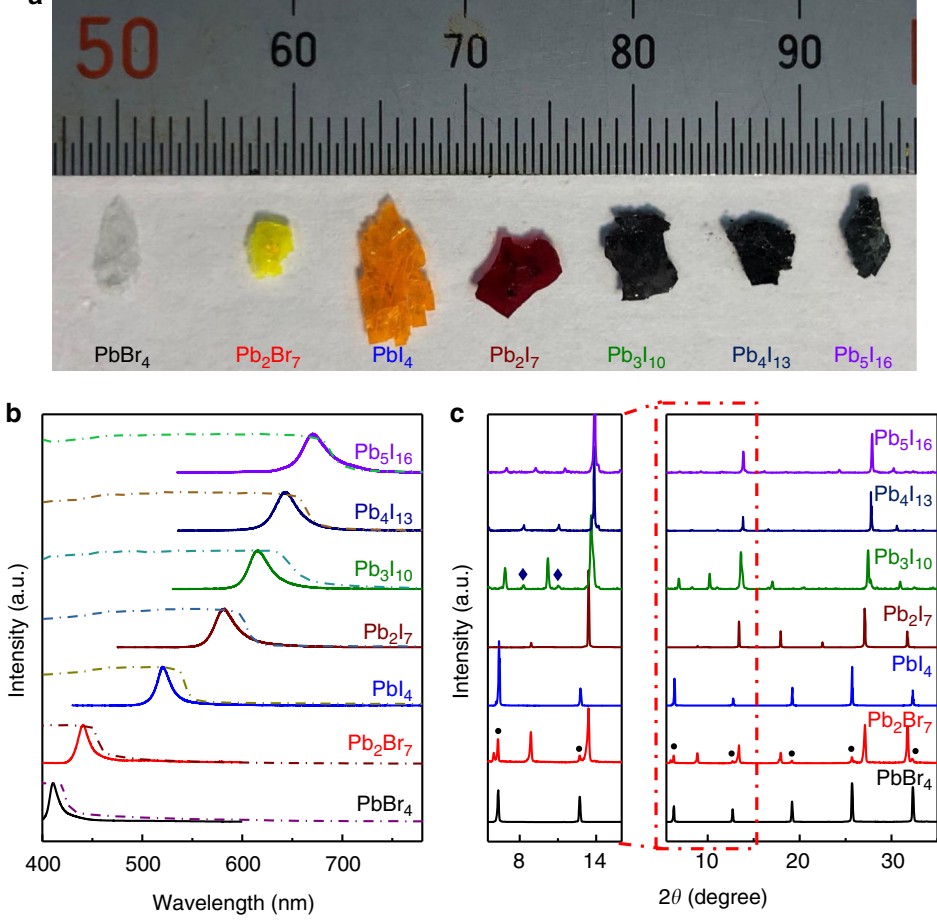

**Fig. 1** Characterizations of 2D perovskite single crystals. **a** Photograph of the as-synthesized perovskite single-crystal plates. From left to right: $(BA)_2(MA)_{n-1}Pb_nBr_{3n+1}$ (denoted as $Pb_nBr_{3n+1}$) and $(BA)_2(MA)_{n-1}Pb_nI_{3n+1}$ (denoted as $Pb_nI_{3n+1}$). **b** Absorption (dash line) and photoluminescence (solid line) spectra of the perovskite plates. **c** Powder XRD spectra of the perovskite crystal plates. The low angle part is magnified to clearly see the diffraction peaks for impurity phases. Black dot: $(BA)_2PbBr_4$ phase within $(BA)_2(MA)Pb_2Br_7$; Navy rhombus: $(BA)_2(MA)_3Pb_4I_{13}$ phase within $(BA)_2(MA)_2Pb_3I_{10}$

monochromatic light illumination. The low dark current of about $10^{-12}$ A, three orders of magnitude lower than that of its 3D counterparts[25], has been observed in our devices. This extremely low dark current can be ascribed to the high crystalline quality of our single crystals with very low concentration of the intrinsic carriers and the extremely low conductivity in the out-of-plane direction[25]. The corresponding dark current density is estimated to be about $10^{-8}$ A cm$^{-2}$, also one order of magnitude lower than that of its 3D counterparts. Under the 620 nm light illumination, the current exhibits a significant increase in contrast to negligible response under a 550 nm excitation, suggesting the excellent wavelength-selective photoresponse. The $I–V$ curve shows a slight nonlinearity, probably due to the different contact of ITO and Au, and the extreme large resistance along the out-of-plane direction, which dominates over the contact resistance. The degree of asymmetry of $I–V$ curves depend on the thickness of the devices with a large asymmetry for thinner devices (Supplementary Fig. 2a). More importantly, no hysteresis appears in $I–V$ curve due to the suppressed ion migration in 2D perovskites[42], which could be beneficial to the long-term stability of the devices.

The optical switch characteristic reveals the excellent stability and reversibility of our devices. The on–off ratio can reach about $10^3$ excited by a 620 nm monochromatic light with a power of 20 μW cm$^{-2}$ (inset in Fig. 2b), larger than the reported value in 3D perovskite cases[12]. The rise and falling time, defined as the

time taken for the photocurrent increasing from 10 to 90% of the peak value and vice versa, were evaluated to be around 100 ms (Fig. 2c). By measuring the response current at different frequencies, the 3 dB cutoff frequency of the device was calculated to be about 20 Hz (Fig. 2d). Fig. 2e presents the EQE of the $(BA)_2(MA)Pb_2I_7$ plate vertical device under different biases. Interestingly, the spectral response exhibits a single narrow peak with the maximum EQE of 300% even under a bias of 10 V, corresponding to an applied electric field of 0.12 V μm$^{-1}$. With the increase of the bias, the EQE shows a gradual increase with the peak position maintaining at the same wavelength. The rejection ratio is calculated to be about 40 in our devices, which is several times smaller than that in 3D perovskite-based narrow-band photodetectors[12]. In particular, such narrowband response has also been observed in a much thinner device of 5 μm under a much larger electrical field of 2 V μm$^{-1}$ (Supplementary Fig. 2).

To evaluate the specific detectivity, we measured the noise current over ten times and calculated the average noise current of our detectors under a bias of 5 V at different frequency (Fig. 2f). The noise current $I_n$ is estimated to be 0.06 pA Hz$^{-1/2}$ at 5 Hz, which is apparently larger than the shot noise limit (3.6 fA Hz$^{-1/2}$) calculated by the formula $I_{s.n} = (2qI_d)^{1/2}$ ($q$ is the electron charge and $I_d$ is the dark current). Noise equivalent power (NEP) was calculated by the formula NEP = $P/(I_s/I_n) = I_n/(I_s/P) = I_n/R$, where $P$ is incident light power intensity, $R$ is the responsivity,

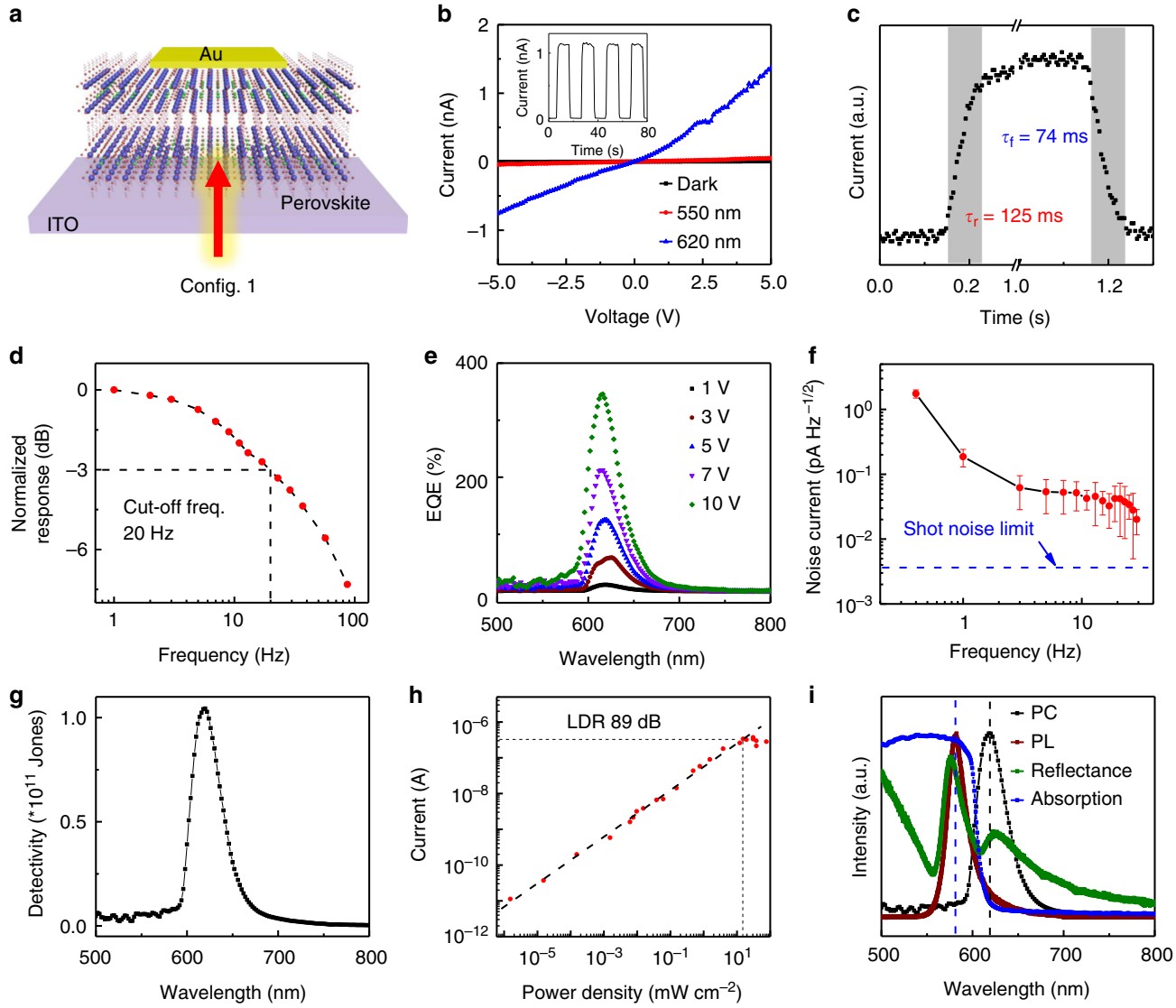

**Fig. 2** Narrowband photodetection based on $(BA)_2(MA)Pb_2I_7$ single-crystal plates. **a** Schematic of the 2D perovskite-based vertical device illuminated from back side (denoted as Config. 1). **b** Dark current and photocurrent of an 80 μm-thick $(BA)_2(MA)Pb_2I_7$ detector under a 550 nm monochromatic light with a power density of 5 μW cm$^{-2}$ (or a 620 nm monochromatic light with a power density of 20 μW cm$^{-2}$). Inset: the optical switching characteristic of the device under a 620 nm monochromatic light illumination with a power density of 20 μW cm$^{-2}$ at a bias of 5 V. **c** Temporal response of the $(BA)_2(MA)$ $Pb_2I_7$ device. **d** Normalized response as a function of modulation frequency. The 3 dB cutoff frequency is around 20 Hz. **e** EQE spectra of the $(BA)_2(MA)$ $Pb_2I_7$ device under different biases. **f** Average total noise measured by lock-in amplifier and preamplifier under a bias of 5 V and calculated shot noise limit (blue dash line). Error bars are the SDs at each frequency. **g** Specific detectivity (D*) spectrum under a bias of 5 V with a modulation frequency of 5 Hz. **h** Linear dynamic range of the detector under a bias of 5 V. **i** Normalized photoconductivity (PC), photoluminescence (PL), reflectance, and absorption spectra of the $(BA)_2(MA)Pb_2I_7$ device

and $I_s$ and $I_n$ are photocurrent and noise current, respectively. NEP is calculated to be 0.1 pW Hz$^{-1/2}$ at the wavelength of 620 nm under a modulation frequency of 5 Hz. Afterwards, the detectivity $D^*$ is calculated by $D^* = (AB)^{1/2}/NEP$ ($A$ is the area of photodetector and $B$ is the bandwidth). The maximum detectivity under a bias of 5 V is evaluated to be about $1 \times 10^{11}$ Jones illuminated by a 620 nm monochromatic light with a power density of 20 μW cm$^{-2}$ (Fig. 2g), about five times larger than that of 3D perovskite-based narrowband photodetectors[12]. Finally, the linear dynamic range (LDR) was measured to be 89 dB by LDR = 20*log ($I_s/I_d$) under a 633 nm laser illumination (Fig. 2h).

This sort of narrowband spectral response has been observed in $(BA)_2PbI_4$ plate vertical devices with the similar performance metric as that of $(BA)_2(MA)Pb_2I_7$ devices (Supplementary Fig. 3). The narrowband spectral response in 2D perovskite plates here

is similar to the reported 3D perovskite-based narrowband photodetectors originated from CCN or surface-charge recombination[12,13]. Nevertheless, compared with 3D perovskite single-crystal narrowband photodetectors, the EQE in our devices is more than two orders of magnitude higher and the measured detectivity is also five times larger[12]. Furthermore, our narrowband photodetectors can function at a much thinner thickness of 5 μm and sustain a much higher electric field of 2 V μm$^{-1}$ compared with their 3D perovskite counterparts (0.5 mm and 0.008 V μm$^{-1}$)[12].

To investigate the origin of the narrowband spectral response and the excellent performance of narrowband photoresponse in our 2D perovskite vertical devices, we have measured the reflection spectra of the devices and plotted together with the absorption, PL, and photoconductivity (PC) spectra in Fig. 2i.

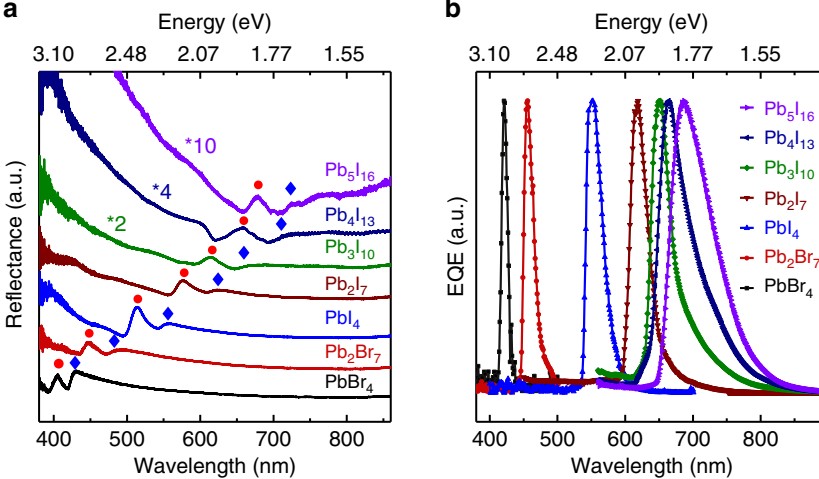

**Fig. 3** Narrowband photodetection in the entire visible range. **a** Reflection spectra of the perovskite single-crystal plates. Red dots indicate the exciton peak, whereas blue rhombus mark the self-trapped states below the bandgap. **b** Normalized EQE spectra of the vertical devices (Config. 1) based on 2D perovskite of different halide compositions and layer number $n$. Measurements were carried out under a bias of 5 V

The PL peak coincides with the absorption edge, suggesting that the emission originates from the free excitonic recombination[37]. Nevertheless, the narrowband spectral response peak in PC spectrum shows a large redshift of 124 meV compared with PL peak, which is far below the excitonic absorption edge (Fig. 2i). The reflection spectrum exhibits an extra peak about 130 meV below the free exciton peak, because the reflection spectrum is very sensitive to the optical response compared with absorption spectrum[43]. The extra reflection peak is close to the narrowband response peak in PC spectrum, implying that the extra reflection peak-associated absorption (denoted as sub-bandgap absorption) likely has a critical role in the narrowband spectral response we observed. The free excitonic absorption and the sub-bandgap absorption make 2D perovskites naturally equip with two absorption onsets, which have two distinctly different absorption regimes with high absorption coefficient for free excitonic absorption and band-to-band absorption, and low absorption coefficient for the sub-bandgap absorption, respectively.

For the high absorption regime, the majority of charge carriers are created in a narrow region close to the surface (termed as surface generation) due to the large absorption coefficient-induced small penetration depth according to the Beer–Lambert law[12]. In contrast, the low absorption coefficient for the sub-bandgap absorption makes the charge carriers be produced in the region deep inside the bulk (termed as bulk generation)[12]. The collection of surface-generated carriers is suppressed via recombination losses due to the possible factors including the imbalanced carriers' transit time, higher local carrier density, and severe surface-charge recombination[12]. Consequently, the photocurrent and EQE at short-wavelength range is bleached due to the suppression of the surface-generated carrier collection. For the bulk-generated carriers, the recombination losses will be much lower leading to a much higher charge collection efficiency and thus a larger photocurrent and EQE. When the wavelengths are far below the exciton absorption peak of the 2D perovskites, the absorption coefficient is reduced further to nearly zero, resulting in the vanished photoresponse. As a result, the combination of the absorption engineering and the charge collection manipulation together leads to the narrowband spectral response of the 2D perovskite-based photodetectors[12]. Therefore, the narrowband spectral response observed in our 2D perovskite device is very likely originated from efficiently controlling the EQE of the devices via the CCN mechanism as previously reported[12].

**Self-trapped states in 2D perovskites**. To verify our hypothesis and further improve the performance of the narrowband photodetection, it is essential to understand the origin of the sub-bandgap absorption peak. One possible origin for such peak is from the self-trapped states. In $(BA)_2(MA)_{n-1}Pb_nI_{3n+1}$ series, previous studies have revealed that self-trapped states are formed locating within around 100–400 meV below the band edge exciton with a broad energetic distribution and weak optical transition strength[29]. The presence of the self-trapped states is supported by the sub-bandgap emission peak at low temperatures, and room-temperature transient and steady-state absorption spectra in $(BA)_2PbI_4$. In particular, the density of the self-trapped states increases when the layer number $n$ decreases from 3 to 1[29].

To confirm that the sub-bandgap absorption peak in the reflection spectrum indeed originate from the self-trapped states in our 2D perovskites, we have carried out the temperature-dependent absorption, and temperature- and power-dependent PL studies (Supplementary Note 1 and Supplementary Figs 4–8). The presence of the self-trapped states in our samples are supported by a series of evidences including the extra emission peaks in PL spectra and sub-bandgap absorption peak (Supplementary Figs 4 to 5), the intensity ratio of self-trapped state-associated reflection peak to the free exciton reflection peak (Supplementary Fig. 6), the strong electron–phonon coupling strength characterized by the Urbach slope (Supplementary Fig. 7), and the power-dependent PL spectra (Supplementary Fig. 8 and Supplementary Note 1). Furthermore, we have also measured the reflection spectra of $(BA)_2(MA)_{n-1}Pb_nBr_{3n+1}$ ($n = 1$ or 2) and $(BA)_2(MA)_{n-1}Pb_nI_{3n+1}$ ($n = 1$ to 5) single crystals (Fig. 3a). As expected, the intensity ratio of the sub-bandgap absorption peak to the excitonic absorption peak gradually decreases with the increase of $n$ number in their respective series due to the increase of the relative surface areas (Supplementary Fig. 6), a strong evidence that the sub-bandgap absorption-associated reflection peak is due to the self-trapped states. The position of sub-bandgap absorption-associated reflection peak also agrees well with the predicated position of self-trapped states[29], which further supports our hypothesis that the sub-bandgap reflection peak originated from the self-trapped states. Furthermore, the long tail of PL spectra also suggests the presence of the self-trapped states in our 2D perovskites, which has been observed in materials with strong electron–phonon coupling (Figs 1b and 2i)[28,44,45].

To exclude the sub-bandgap reflection peak and PL tailing from permanent defects, we have performed the low-temperature PL (Supplementary Fig. 8a) and power-dependent PL (Supplementary Fig. 8b). The PL tailing splits into several emission peaks and these peaks show a linear dependence with the excitation power. The PL peaks show no shift with the excitation power, which suggest the additional peak are not from permanent defects (Supplementary Note 1). Therefore, based on the above discussions, we believe the sub-bandgap reflection and emission peaks originated from the self-trapped states.

**Narrowband photoresponse in entire visible range**. The generic presence of two absorption onsets due to the excitonic absorption and the self-trapped states in $(BA)_2(MA)_{n-1}Pb_nBr_{3n+1}$ ($n = 1$ to 2) and $(BA)_2(MA)_{n-1}Pb_nI_{3n+1}$ ($n = 1$ to 5) series enable us to fabricate narrowband photodetectors based on such 2D perovskite series in the entire visible spectral range. Fig. 3b displays the normalized EQE of devices with 2D perovskite single crystals of different halide compositions and layer number $n$. To compare among different devices, we intentionally selected the 2D perovskite single-crystal plates with thickness around 80 μm and a bias of 5 V. Importantly, only one single narrow peak was observed in the EQE spectra of all devices based on $(BA)_2(MA)_{n-1}Pb_nBr_{3n+1}$ ($n = 1$ to 2) and $(BA)_2(MA)_{n-1}Pb_nI_{3n+1}$ ($n = 1$ to 5) series with the response peak overlapping with the self-trapped state-associated reflection peak, which further verifies that the self-trapped states is crucial for the narrowband spectral response (Supplementary Fig. 9). The FWHM of all devices are < 60 nm and strikingly the FWHM can be as narrow as 10 nm around 420 nm (Table 1), whereas EQEs are on the same order of 200% for all devices. With the increase of $n$, although the intensity of self-trapped state-assisted absorption is weakened, the conductivity in the out-of-plane direction nevertheless continuously increases[25], which would be beneficial to the carrier extraction. Those two factors together result in similar EQE for all $(BA)_2(MA)_{n-1}Pb_nI_{3n+1}$ crystals with different $n$. The gradually broadening of the FWHM with the increase of the layer number $n$ might be partially ascribed to the presence of small inclusions of the hybrid perovskite phases with different $n$ number for $(BA)_2(MA)Pb_2Br_7$ and $(BA)_2(MA)_{n-1}Pb_nI_{3n+1}$ ($n = 3$ to 5). It is expected that the FWHM can be further narrowed by improving the crystalline quality and phase purity of 2D perovskites. This tunable narrowband spectral response across the whole visible range is promising for many practical applications in the field of imaging and machine vision.

**Mechanism of narrowband spectral response**. To further confirm the mechanism of the narrowband spectral response is the CCN concept together with the self-trapped states, we have fabricated the $(BA)_2(MA)Pb_2I_7$ plate-based lateral devices on glass substrate illuminated from both back (Config. 2) and front side (Config. 3) (Fig. 4a). Fig. 4b represents the normalized EQE spectra of all three different measurement configurations. For the back-illumination measurement configuration (Config. 3), the similar narrowband spectral response was observed with the response peak locating at nearly the same position compared with the vertical devices (Config. 1). Nonetheless, for the front-illumination measurement configuration, a broadband spectral response was observed with three response peaks, which are labeled as $B$, $X_0$, and $X_t$. When the EQE spectrum of Config. 3 was compared with the PL, absorption, and reflection spectra (Supplementary Fig. 9), we can assign the $B$ peak to band-to-band transition, $X_0$ to free exciton peak, and $X_t$ to the self-trapped state-associated absorption. Moreover, the $X_t$ peak matches well with the narrowband response peak observed in Config. 1 and Config. 2, which can further prove that the narrowband response

**Table 1 Summary of FWHM and peak positions of the narrowband photodetectors extracted from EQE spectra shown in Fig. 3b**

| Perovskite | FWHM (nm) | Peak position (nm) | Thickness (μm) |
|---|---|---|---|
| $PbBr_4$ | 10 | 420 | 60 |
| $Pb_2Br_7$ | 17 | 455 | 90 |
| $PbI_4$ | 30 | 550 | 80 |
| $Pb_2I_7$ | 30 | 620 | 80 |
| $Pb_3I_{10}$ | 35 | 650 | 50 |
| $Pb_4I_{13}$ | 55 | 665 | 80 |
| $Pb_5I_{16}$ | 60 | 690 | 80 |

is due to the self-trapped states rather than the free exciton absorption.

It should be noted that the EQE of the $X_t$ peak is comparable to that of $B$ and $X_0$ peaks as expected due to the self-trapped state-enhanced absorption in the sub-bandgap regime. The exact same measurement with the same device configurations has been carried out for $(BA)PbI_4$ devices and similar results have been obtained, suggesting that the narrow spectral response in 2D perovskite series has the common underlying mechanism (Supplementary Fig. 10). Furthermore, the normalized EQE spectra show the $X_0$ peak exhibits a slightly larger slope than that of $B$ peak with the increase of the bias, indicating that the external electric field only contributes minor to the exciton ionization (inset of Fig. 4c). In contrast, with the increase of the bias, the trend of $X_t$ peak gradually deviates away from that of the $X_0$ and $B$, further verifying that $X_t$ has different origins from $X_0$ and $B$.

Based on the above experimental results, we schematically illustrate the carrier generation and extraction process in all those three different measurement configurations in Fig. 4d–f, respectively. In the vertical devices (Config. 1, Fig. 4d), the surface-generated carriers (corresponding to the short-wavelength light) suffer severe recombination losses as mentioned above and thus has a lower collection efficiency, whereas the bulk-generated carriers (corresponding to long-wavelength light below bandgap energy) can be efficiently collected with a high efficiency, leading to a narrowband spectral response. This is exactly the CCN concept used to achieve narrowband photodetectors in 3D perovskites and other organic semiconductors[12,13]. For the lateral devices with the back illumination (Config. 2), the carrier generation process is exactly the same as that in Config. 1, except that the direction of external electrical field is different, which would affect the carrier extraction efficiency (Fig. 4e). Therefore, narrowband spectral response is also anticipated in Config. 2. Nevertheless, for the lateral devices with front illumination (Config. 3), both the surface-generated carriers and bulk-generated carriers can be collected by electrodes, resulting in the broadband spectral response (Fig. 4f). However, due to the different carrier collection efficiency caused by the different carrier transport pathway between the surface-generated carriers and bulk-generated carriers, the EQE of the $B$ peak, $X_0$ peak, and $X_t$ peak show different bias dependences as shown in inset of Fig. 4c.

The proposed carrier generation and extraction processes have been further verified by the thickness-dependent EQE spectra in $(BA)_2(MA)Pb_2I_7$ plates. In both the vertical devices (Config. 1) and lateral devices with back illumination (Config. 2), the narrowband response peak $X_t$ gradually moves toward and finally merges into the exciton peak $X_0$ when the thickness of the single-crystal plates is reduced down to a certain value (e.g., 1.4 μm for the vertical devices and 10 μm for back-illuminated lateral devices), when the broadband spectral response takes place (Fig. 4g, h). The blueshift of the narrowband response peak $X_t$

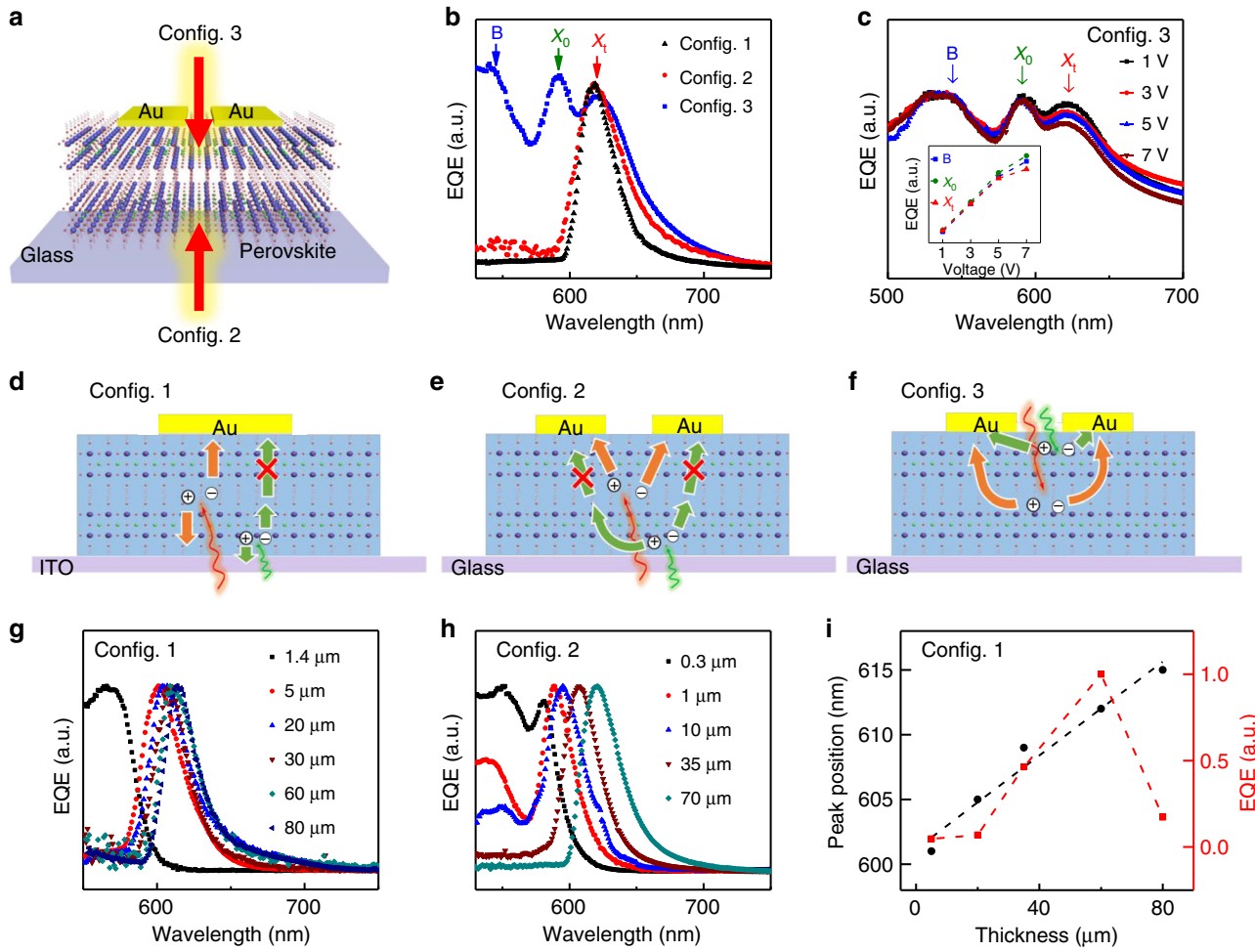

**Fig. 4** Mechanism study of the 2D perovskite based narrowband photodetection. **a** Schematic of the 2D perovskite single crystal-based lateral device with illumination from back side (Config. 2) and front side (Config. 3). **b** Normalized EQE spectra of the $(BA)_2(MA)Pb_2I_7$ devices with three different measurement configurations. The peaks of the spectra are marked by arrows with different color. Blue arrow: band-to-band ($B$); green arrow: free exciton ($X_O$); red arrow: self-trapped state ($X_t$). **c** Normalized EQE spectra of device with Config. 3 under different biases. Inset: the dependence of the EQE value of $B$ peak, $X_O$ peak, and $X_t$ peak on bias. The dashed lines are used to guide the eye. **d,e,f** Schematic of the carrier generation and extraction process for the above mentioned three different measurement configurations. Red: long wavelength light and bulk-generated carriers. Green: short-wavelength light and surface-generated carriers. **g,h** The thickness dependence of normalized EQE spectra of the $(BA)_2(MA)Pb_2I_7$ devices for Config. 1 (**g**) and Config. 2 (**f**), respectively. **i** Thickness dependence of peak position under the same applied electrical bias and normalized EQE under the same applied electrical field extracted from the EQE spectra in **g**. The dashed lines are used to guide the eye

with thickness is resulted from the competition between the recombination losses and charge collection efficiency. Although the longer wavelength light is able to penetrate deeper inside the plates leading to a reduced recombination loss and thus a higher EQE provided that the absorption coefficient change can be neglected, the carriers generated deeper inside the plates would reduce the charge collection efficiency. As a consequence, the synergistic effect of the recombination losses and charge collection efficiency finally determines the peak position of the narrowband spectral response[46]. By reducing the thickness of the 2D perovskite plates, the larger electric field in the thinner devices under the same bias can push the photogenerated carriers away from the surface and thus be able to contribute the current for the vertical devices (Config. 1 and Fig. 4g). As a consequence, the broadband spectral response in thinner devices was observed, which further supports that the narrowband response is a bulk effect via CCN mechanism rather than a surface effect[13].

For the lateral devices illuminated from back side (Config. 2), when the thickness is reduced to 10 μm, the band-to-band peak $B$

starts to appear and the EQE of $B$ peak increases with further reducing the thickness (Fig. 4h). Close inspection of the EQE spectra of a 10 μm-thick device reveals that the $X_t$ peak consists of two peaks corresponding to the $X_O$ and $X_t$ peak, which merge into one broad peak under a higher bias due to the electrical field induced peak broadening (Supplementary Fig. 11). Further reducing the thickness to 0.3 μm, the $X_t$ peak is no longer observed, as the surface-generated carriers dominate over the bulk-generated carriers due to weak absorption of self-trapped states and thus CCN cannot be achieved, leading to the broadband spectral response and the absence of $X_t$ peak. All evidences mentioned above point that the narrowband photodetections in 2D perovskites are due to the CCN mechanism together with the self-trapped state-enhanced sub-bandgap absorption.

Furthermore, the conductivity anisotropy in 2D perovskites can lead to different charge extraction efficiency along the in-plane and out-of-plane direction, and thus the different thickness-dependent narrow spectral response peak position for different device configurations. Indeed, the narrowband response

peak in the lateral devices with back illumination (Config. 2) is more sensitive to the thickness compared with that of the vertical devices (Config. 1) (Fig. 4i and Supplementary Fig. 12). The narrowband response peak can be tuned more than 20 nm by changing the thickness of the plates (Fig. 4i), which provides an extra degree of freedom to extend the narrowband spectral response range. Compared with the monotonic shift of the response peak with the thickness, the EQE at peak position under the same electric field first increases and then decreases with decreasing of the thickness (Fig. 4i) in contrast to that in 3D perovskites[46] (Supplementary Note 2 and Supplementary Fig. 13), which further confirms that the self-trapped states have an indispensable role in the narrowband spectral response. The absorption dip between the free exciton absorption and the self-trapped state-associated absorption peak (Supplementary Fig. 5) leads to a reduced absorption for plates with thickness below a certain value (e.g., 60 μm) when the narrowband response peak moves toward the absorption dip, resulting in a decreased EQE. For the very thick devices (e.g., 80 μm), the low-charge collection efficiency starts to dominate the EQE, leading to a decrease of the EQE again. As a consequence, the observed EQE shows non-monotonic behavior with the thickness of 2D perovskite plates.

## Discussion

The high performance in our narrowband photodetectors compared with that of 3D perovskite-based ones can be attributed to the enhanced sub-bandgap absorption by self-trapped states and large conductivity anisotropy along the in-plane and out-of-plane direction, which is supported by the fact that the EQE of the vertical devices (Config. 1) is one order of magnitude smaller than that of the lateral devices with back illumination (Config. 2) under the exact same measurement conditions (Supplementary Fig. 14). The presence of self-trapped states can enhance the sub-bandgap absorption leading to the high EQE in our narrowband devices. The extremely small conductivity along the out-of-plane direction in 2D perovskites due to the presence of the organic layers can efficiently suppress the dark current ($10^{-12}$ A) and enhance the on–off ratio ($10^3$), as well as significantly reduce the device thickness (5 μm) and sustain a larger electrical field (2 V μm$^{-1}$) with narrow FWHM below 60 nm in the 2D perovskite-based narrowband photodetectors. The narrowband spectral response has been achieved in 2D $(BA)_2(MA)Pb_2I_7$ vertical devices (Config. 1) with a thickness even as small as 5 μm and applied electrical field of 2 V μm$^{-1}$, around two orders of magnitude thinner than those in 3D perovskite single crystal-based narrowband photodetectors[12,46,47] (Supplementary Table 1). More importantly, the 2D perovskite-based narrowband photodetectors inherit the great environmental stability of 2D perovskite compared with their 3D counterparts. Being stored under ambient condition for 45 days, no noticeable performance degradation of the 2D perovskite narrowband photodetectors was observed (Supplementary Fig. 15).

The self-trapped states in 2D perovskites strongly depend on the organic spacers, which leads to the different electron–phonon coupling strength and thus the degree of the lattice distortion. Although strong broadband white light emission has been observed for some sorts of 2D perovskites, self-trapped states could give rise to multiple emission peaks with relatively narrow bandwidth or weak emission peak in other sorts of 2D perovskites[48]. In our $(BA)_2(MA)_{n-1}Pb_nI_{3n+1}$ series, two or three emission peaks (ST1, ST2, and/or ST3) originated from self-trapped states are observed, whereas ST1-associated absorption peak shows a stronger absorption spectrum than ST2 and ST3. ST2 and ST3 only presented as a long tailing due to the rather

weak optical transition strength (Supplementary Fig. 5). In our devices, ST1 absorption with a relatively narrow bandwidth is believed to be responsible to the sub-bandgap photoresponse and thus we can achieve the narrowband photodetection assisted by the self-trapped states.

Similar to 3D perovskite case, the regular band-tail states would expect to also lead to the narrowband photoresponse in 2D perovskite single crystals. Nevertheless, without the contribution from self-trapped states, only one broadened free exciton peak should be present in Config. 3 and the EQE of the narrowband response would continuously decrease with the increase of the thickness of the samples as 3D perovskite case (Supplementary Fig. 13). Rather, we have observed a distinctive $X_t$ peak more than 120 meV below $X_0$ peak (Fig. 4b, c) and the EQE in our 2D perovskite devices first increases and then decreases with thickness (Fig. 4i), both of which strongly suggest that the self-trapped states have an indispensable role to the narrowband response in 2D perovskites. Furthermore, we expect that the EQE of 2D perovskite-based devices would be much smaller than that of 3D counterparts if the regular band-tail states lead to the narrowband photoresponse, as the large resistance in the out-of-plane direction makes it more difficult for carriers to be extracted. In contrast, we observed an enhancement of EQE in 2D perovskite-based narrowband photodetectors, which should result from the enhanced absorption due to the presence of self-trapped states. All those evidences support that the self-trapped states should be responsible to the narrowband response in our devices (Supplementary Note 2).

Finally, it should be noted that the exciton binding energy in 2D perovskites is usually hundreds of meV and thus excitons cannot be ionized by thermal excitation at room temperature[49]. Nevertheless, we have observed strong exciton peak in the PC spectra (Fig. 4b, c), which is unexpected, as excitons are charge neutral and cannot contribute to the photocurrent unless they are ionized. The applied external electric field in our devices is not large enough to ionize the excitons either. It is likely to be that surface depletion field is responsible for the exciton ionization here[50]. The thickness dependence room-temperature PL measurement reveals the exciton emission has been significantly quenched in a thinner plate, suggesting the exciton ionization near the surface (Supplementary Fig. 16). Due to the large absorption coefficient for free excitons, the excitons are mainly generated near surface and subsequently the majority of them are ionized by the surface depletion field, resulting in the photocurrent, which also supported by the fact that the EQE of $X_0$ peak is comparable to that at band-to-band transition peak B (Fig. 4b, c). In terms of the self-trapped states, previous studies have showed that those self-trapped states are excitonic and thus a self-trapped exciton is formed by trapping an electron and a hole[51]. As the self-trapped excitons are generated inside the plates, surface depletion field is unlikely to be responsible for the self-trapped exciton ionization. One possibility is that the bulk excitons are separated at the interfaces between the organic layers and inorganic layers, and then collected by electrodes under external applied electric field, leading to the photocurrent.

In summary, we have demonstrated narrowband photodetections based on 2D perovskite single-crystal plates across the entire visible spectrum with excellent performance by tuning the halide compositions and the layer number $n$. The highly narrowband spectral response can be attributed to the self-trapped states and extremely large conductivity along the out-of-plane direction. Compared with 3D counterparts, our 2D perovskite-based narrowband photodetectors show a greatly enhanced EQE due to the enhanced absorption in the sub-bandgap regime and can sustain two orders of magnitude higher electrical field and be made two orders of magnitude thinner. Our work provides a simple strategy

to achieve filterless narrowband photodetector in the entire visible wavelength range with high performance for a variety of applications including imaging and sensing.

## Methods

**Sample preparations**. The synthesis of 2D perovskite single crystals is exactly same as previously reported.[34,35] To synthesize the methylammonium chloride (MACl) solution, 40% w/w aqueous methylamine (MA) was first mixed with 32 % w/w aqueous hydrochloric acid at a molar ratio of 1:1 under stirring at 0 °C for 2 h. Then the solvent of the resultant solution was evaporated at 60 °C followed by washing with diethyl ether and subsequent drying at 70 °C for 12 h. The same procedure was utilized to synthesize $n$-butylammonium iodide (BAI) solution, except that $n$-butylamine (BA) was used to substitute MA and the solution was stirred for a longer time of 4 h. For the synthesis of $(BA)_2(MA)_{n-1}Pb_nI_{3n+1}$ ($n = 1$ to 5) single crystals, the precursor solution was prepared by dissolving 0.5 g PbO powder into a mixture of 3 ml 57% w/w aqueous hydriodic acid (HI) solution and 0.5 ml hypophosphorous acid ($H_3PO_2$, 50 wt% in water) solution by heating to 140 °C under constant stirring. $(BA)_2PbI_4$ crystals were synthesized by injecting 2.5 mmol BAI solution into the precursor solution, whereas BAI solution was replaced by a mixed solution of (1.25 mmol; 1.67 mmol; 1.86 mmol; 2 mmol) MACl with (1.75 mmol; 0.83 mmol; 0.62 mmol; 0.5 mmol) BAI for the synthesis of $(BA)_2(MA)_{n-1}Pb_nI_{3n+1}$ ($n = 2$ to 5) plates. Then, the solution naturally cooled down to room temperature. To complete the growth, the resultant precipitation was left overnight. The procedure of synthesis of $(BA)_2PbBr_4$ and $(BA)_2(MA)Pb_2Br_7$ are exactly the same as that of synthesis of $(BA)_2PbI_4$ and $(BA)_2(MA)Pb_2I_7$, except using a hydrobromic acid (HBr, 48 wt% in water) solution instead of HI solution.

**Fabrication of ITO (glass)/perovskite/Au devices**. The ITO (glass) substrate was cleaned with isopropanol via ultrasonic process and dried by pure nitrogen, which was then treated by oxygen plasma for 3 min. Perovskite plates with different thicknesses were exfoliated onto the ITO (glass) substrate. Subsequently, a gold foil was placed on perovskite plate under the aid of an optical microscope (Olympus BX53).

**Fabrication of two-probe devices**. The two-probe lateral devices were fabricated by directly placing the exfoliated microplates onto the predefined electrodes (Supplementary Fig. 17a). Five nanometers of Cr/50 nm Au electrodes were defined by photolithography and deposited by thermal evaporation onto 300 nm $SiO_2$ on Si wafer. Then the perovskite microplates were exfoliated from respective bulk crystals onto polydimethylsiloxane (PDMS) and an optical microscope coupled with a micromanipulator was used to position the exfoliated microplate onto the predefined electrodes.

**Fabrication of Au/perovskite/graphene devices**. We adopted aligned and transferred technique previously developed[36] to fabricate the vertical device with Au as a bottom contact and graphene as a top contact (Supplementary Fig. 17b). First, the perovskite microplates were exfoliated from the bulk crystals onto PDMS and an optical microscope coupled with a micromanipulator was used to put the exfoliated microplates on the predefined electrodes. Then a few-layer graphene flake was aligned and transferred under the help of optical microscope and micromanipulator to bridge the pre-transferred perovskite and another electrode by using the same technique previously reported[36]. Finally, the PMMA (polymethyl methacrylate) film was dissolved by chloroform.

**Material characterizations**. The PL measurements were performed on a home-built Raman spectrometer (iHR-550 Horiba) with a 600 g mm$^{-1}$ grating, whereas a 473 nm solid-state laser or a 325 nm He-Cd laser was used as the excitation source. The reflection measurements were carried out on the same setup using a halogen lamp as the light source. XRD patterns were measured on a Bruker D2 PHASER (Cu Kα $\lambda = 0.15419$ nm, Nickel filter, 25 kV, 40 mA). Absorption spectra were performed on an UV-Vis spectrophotometer (UV-1750, SHIMADZU). Thickness of the crystal plates was measured by stylus profilometry (Bruker DektakXT) or atomic force microscope (Bruker Dimension Edge).

**PC measurement**. The PC was measured in a home-built PC system. We used a halogen lamp as the light source, which was dispersed by a monochromator (Horiba JY iHR320). The monochromatic light output was collimated to the device and a lock-in amplifier (Stanford SR830) was used to collect the photocurrent, while a mechanical chopper (Stanford SR 540) provides the reference signal for the lock-in amplifier. Both the noise and frequency modulation measurements are carried out by coupling a low-noise preamplifier (Stanford SR 540) with a lock-in amplifier (Stanford SR830). All filters of preamplifier have been switched off for the noise measurement to avoid the influence from those filters. The output ($I$–$V$) curves were measured by a DAQ-card (National Instrument PCI 6030E) coupled with a low-noise current preamplifier (Stanford SR570). The transient photocurrent signal was acquired by a digital oscilloscope (Tektronix MDO3032). The

LDR was measured by using a 633 nm He-Ne laser coupled with a series of neutral density filters to tune the light intensity. All measurements were performed at ambient condition and room temperature.

## Data availability

The data that support the findings of this study are available from the corresponding author upon reasonable request.

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

## Acknowledgements

D.L. acknowledges the support from NSFC (61674060) and the Fundamental Research Funds for the Central Universities, HUST (2017KFYXJJ030, 2017KFXKJC003, 2017KFXKJC002, and 2018KFYXKJC016). We thank the Testing Center of Huazhong University of Science and technology for the support in X-ray diffraction measurement and thank the Center of Micro-Fabrication of WNLO for the support in device fabrication.

## Author contributions

D.L. conceived the idea and guided the project. J.L. performed most of the experiments. J. W. and J.M. synthesized perovskite crystals. H.S. contributed to the devices fabrications and L.L. assisted in photoconductivity measurements. J.L., D.L., and X.D. wrote the paper. All authors discussed the results and commented on the manuscript.

## Additional information

**Competing interests:** The authors declare no competing interests.

