## [Peer Review File · Nature Communications]

Editorial Note: Parts of this peer review file have been redacted as indicated to remove third-party material where no permission to publish could be obtained.
Parts of this peer review file have been redacted as indicated to maintain the confidentiality of unpublished data.

Reviewers' comments:

Reviewer #1 (Remarks to the Author):

The authors reported filterless narrowband photodetectors based on 2D Ruddlesden-Popper layered perovskites. They fabricated single crystals of 2D perovskites with controlled layer numbers and compositions. Based on these perovskite single crystals, they realized narrowband photodetection with broad spectral tunability. Compared to their 3D counterparts, 2D perovskites sustain narrowband response with reduced device thickness. Details are as follows.

1. The authors attributed the narrowband photodetection to the self-trapped states. However, I cannot find any solid evidence on the presence of self-trapped excitons. The only evidence presented in this manuscript is redshift peaks in the reflection spectra. However, these additional peaks (Fig. 3a) can be caused by the domains with higher layer numbers, edge states and permanent defects. The PL tails (Fig. 2f) can also originate from these factors. Actually, according to the published results, I think the exciton self-trapping does not occur in BA 2D perovskites. The systems with self-trapped excitons usually feature broad, Stokes-shifted PL emission and relatively long carrier lifetime. The interactions between the exciton and deformable lattice will induce the loss of exciton energy and elongation of carrier lifetime. The single crystals of BA layered perovskites exhibit narrow PL emission, small Stokes shift and short lifetime of less than 10 ns (Science 355, 1288-1292 (2017)). However, I can accept the concept of self-trapped states if authors can perform more rigorous experiments. For example, the Urbach edge slope coefficient extracted by the temperature-dependent absorption can be employed to identify the exciton self-trapping (K.S. Song, R.T. Williams, Self-Trapped Excitons, 1996, Springer).

2. Even without the concept of self-trapped states, the narrowband photodetection can be explained. The authors can consider the limited exciton diffusion length induced by the organic hopping barriers in 2D perovskites. With limited diffusion length, the excitons localized on the crystal surface, which generated by high-energy photons, are difficult to diffuse onto electrodes before recombination in config. 1 and 3 devices. To evaluate this mechanism, the authors can measure the exciton diffusion length in single crystals of 2D perovskites.

3. The authors shows EQEs of exceeding 200 % in 2D-perovskite photodetectors. For a device with photodiode configuration, this represents a large photoconductive gain. The authors should carefully study and discuss the gain mechanism.

4. The authors show nearly symmetric IV curve in Fig. 2b. However, the device configuration is photodiode with different contacts of ITO and Au. The presence of Schottky barrier should lead to the asymmetry of IV curves. The authors need to carefully check their data or provide some explanation.

5. The XRD peaks ranging from 2θ to 15θ are very important to identify the layer numbers in 2D perovskites. However, in Fig. 1c, these peaks is not presented clearly, especially for $n=4, 5$ perovskites. The $n=3$ perovskites contain some $n=4$ phase, which was not mentioned in manuscript. The authors need to pay more attention to the preparation of 2D perovskites with pure phase (Pb₂Br₇ and Pb₃I₁₀). The phase purity is important for the study of physical properties and the photodetectors with narrower spectral response.

6. The FWHM and peak position of spectral response of photodetectors are dependent on the thickness of crystals. However, I cannot find any information about the crystal thickness in Table 1. The authors are required to study the FWHM and peak positions on each 2D perovskites with different thicknesses, instead of providing values without sample information.

7. Fig. 4i and Fig. S6 provide thickness-dependent peak positions. The authors seemed to fit these data points by a linear function. What is the physical model used in these fitting?

8. This manuscript provides a relatively high detectivity of 10^{13} Jones. The authors should provide the value of light power for calculating this detectivity. Additionally, power-dependent photocurrents and responsivities are required. The linear dynamic range of these photodetectors need to be calculated.

9. The detectivity was calculated by shot noise limit and steady-state photocurrents. Therefore, this value was overestimated. The authors need to rigorously calculate the specific detectivities by measuring the noise currents.

In summary, I cannot recommend this manuscript for the publication of Nature Communication at the present stage. However, I think I can reconsider the revised manuscript on condition, if authors could provide rigorous demonstration on the self-trapped states and could added systematic measurements on the figures of merit (spectral response, power-dependent photocurrents and noises) of devices.

Reviewer #2 (Remarks to the Author):

In their manuscript "Self-trapped states enabled filterless narrowband photodetections in 2D layered perovskite single crystals" Li et al report the filterless narrowband 2D perovskite single-crystalline photodetectors enabled by self-trapped states mechanism. The device performance is very promising compared to previous 3D perovskite narrowband photodetectors and brings new insights into applications of 2D perovskite multi-quantum well single crystals. This study may sprout future researches on 2D halide perovskite monocrystalline photodetectors and benefit broad scientific community. Nevertheless, the "Self-trapped states" mechanism proposed by the author remains arguable. The authors are suggested to provide additional evidences and proper physical interpretation illustrating such a mechanism. Besides, there are a few issues the authors should also consider addressing.

Some sentences are in bold with no reason, e.g., line 88th-92nd in page 4.

Page 6, line 119-120, "...which can be attributed to the reduced quantum confinement effect for larger n number..." is not strictly correct. The bandgap evolution is a result of many-body interaction and exciton binding energy change through both quantum and dielectric confinement upon layer number n increase. The author is suggested to give a complete statement.

Figure 1b, all the samples exhibit blueshift of PL peak from their absorption edge. While typically the photon energy measured via PL is lightly smaller than the optical bandgap obtained from the absorption edge by an amount of the binding energy, that should be redshift. The author needs to provide some information or discussions based on this concern. Figures such as Stokes shift as a function of n might be helpful.

Page 8, line 165-167, "...the external quantum efficiency in our devices is more than two orders of magnitude higher than that in 3D perovskite single crystal narrowband photodetectors..." The EQE can be magnified significantly through enlarging the driving voltage through multiple mechanisms. In the cited reference of 3D perovskite single crystal narrowband photodetectors (Nature Photonics 9, no. 10 (2015): 679-686), the driving voltage is 4 V and an electric field of ~ 0.01 V/ μm can induce a 20% EQE. Then the 400% EQE in this report (under a bias of 10 V, corresponding to an applied electric field of 0.12 V/ μm) seems not a big enhancement.

Page 9, in authors' assumption, the narrow photoresponse comes from the "low absorption coefficient of the sub-bandgap absorption" as it provides bulk-generated carriers that are hard to recombine. Then the sub-band states are majorly responsible for the photoresponse. Since single crystals are relatively pure with less sub-band states compared to polycrystals, as those single crystal devices

perform quite well, how about the polycrystal devices? In addition, the self-trapped states in white LED the author cited in Page 4 have wide distributions so that the emitting light span a wide range of wavelength. Interestingly, the sub-band states assumption in this study delivers a narrow photoresponse. The author needs to give some clarifications to make the consistence with previous reports.

Page 9, line 196,197, "...When the wavelengths are far below the optical gap of the 2D perovskites, the absorption coefficient decreases further to nearly zero and thus the photoresponse can be neglected ..." it is confusing why absorption coefficient decreases to zero at far short wavelength region, as in the absorption spectrum, the absorbance keeps invariant in the region below optical bandgap.

Page 10, line 218, typo "blow-gap".

Reviewer #3 (Remarks to the Author):

This manuscript by Li et al. reported the filterless narrowband photodetector based on 2D layered perovskite single crystal. The authors did a comprehensive work on identifying the effect of the self-trapped states, which is unique to the 2D layered perovskite, on the device operation mechanism of the narrowband photodetectors. However, the basic working principle of the device is still the charge collection narrowing induced narrowband photon response which is not new. Moreover, I remain suspicious of whether the self-trapped states is indeed good for narrowband photodetection, considering the board energetic distribution of these below gap states that will inevitably increase the Urbach energy and thus giving rise to a large FWHM of the photon response spectrum. In fact, the FWHW of the current narrowband photodetector based on 2D perovskite single crystal is still considerably larger than that of the photodetector made from 3D perovskite single crystal reported previously (see Ref. 12). In addition, the authors claimed orders of magnitude higher sensitivity of the 2D perovskite single crystal based photodetectors compared to the 3D perovskite based ones, which unfortunately is based on a wrong characterization method. They assumed the shot noise from dark current is the major contributor to the total noise, which however is not true in most cases. I suggest the authors directly measure the noise, and do the linear dynamic range characterization to verify the high sensitivity of their photodetectors. Therefore, in my opinion this work may be more suitable for a specialized journal after revisions, but lacks sufficient novelty and context for a high profile journal like Nature Communications.

Some typos:

Page 10, "blow-gap" should be "below-gap";

Page 16, "CNN concept" should be "CCN concept".

Reply to reviewers:

We sincerely thank the reviewers for their detailed reading and specific comments which are very important for us to improve the manuscript. The reviewers mainly concern about whether there are self-trapped states in our 2D perovskites and whether the self-trapped states play a role in the narrowband photoresponse. The reviewers also question the accuracy of the detectivity we measured. To avoid repetition, we start with some general statements to address those concerns one by one and thereafter the point-to-point response to the reviewers is provided.

1. The presence of the self-trapped states in our 2D perovskites.

The formation of the self-trapped states is due to the strong electron-phonon interaction, which would introduce the lattice distortion and thus trap carriers and/or excitons within the potential wells. As has mentioned in our original manuscript, theoretical studies have predicted that the self-trapping critically depends on the dimensionality of the systems. In three-dimensional (3D) case there is a potential barrier for the self-trapping while no such barrier is present for quasi-one-dimensional systems. Two-dimensional (2D) systems are marginal cases with a much lower potential barrier or nonexistent potential barrier, leading to the easier formation of self-trapped states [K.S. Song, R.T. Williams, *Self-Trapped Excitons*, 1996, Springer]. In 2D perovskites, previous studies revealed that this enhanced self-trapping introduces the extra absorption and emission peak below bandgap and thus greatly alter the electrical and optical properties of the 2D perovskites including the thermal activated broad white light emission [J. Phys. Chem. C. 119, 23638–23647 (2015)] and hopping transport of charge carriers [J. Phys. Chem. Lett. 9, 1434-1447 (2018)]. Also, for some sorts of 2D perovskites, multiple broad emission peaks from self-trapped states have been observed originated from self-trapping of electrons, self-trapping of holes and/or self-trapping of both electrons and holes, which gives the different trapping energies and thus different emission peaks (Fig. R1a and b, [J. Mater. Chem. C, 5, 2771-2780. (2017)]). In $(\text{BA})_2(\text{MA})_{n-1}\text{Pb}_n\text{I}_{3n+1}$ series, the self-trapped states have been also observed in previous report, supported by the sub-bandgap emission peak at low temperatures and room-temperature transient and steady-state absorption spectra (Fig. R1c-e, [J. Am. Chem. Soc. 137, 2089–2096 (2015); ACS Nano. 11, 10834–10843 (2017)] and [Nature Communications. 9, 2254 (2018)]). In the following, we will provide more evidences that the self-trapped states are indeed formed in our 2D perovskite series.

[Redacted]

Fig. R1. (a) Temperature-dependent PL spectrum of (EDBE)PbBr₄. W1, W2, W3 indicate three principal components originated from different trapped-states. **(b)** Schematic representation of the emissive process in the white-light generation of **(a)**. **(a)** and **(b)** are taken from [J. Mater. Chem. C, 5, 2771-2780. (2017)]. **(c)** Normalized absorption and PL spectra of (BA)₂PbI₄ and (HA)₂PbI₄ polycrystalline thin films. (c) is taken from [ACS Nano. 11, 10834–10843 (2017)] **(d)** Temperature-dependent PL spectrum and **(e)** transient absorption spectrum of (BA)₂PbI₄. **(d)** and **(e)** are taken from [J. Am. Chem. Soc. 137, 2089–2096. (2015)].

First, we have carried out the temperature dependent micro-photoluminescence (μ -PL) and micro-absorption measurements on the exfoliated microplates to check whether there are indeed self-trapped states in our samples. We intentionally used exfoliated microplates with thicknesses of \sim 100 nm so that the absorption would not saturate at the free exciton band and above the bandgap regime. Under such case, the weak self-trapped state induced absorption might be observed. In bulk crystals, **the absorbance can rapidly change over several orders of magnitude near the free exciton absorption peak and thus the weak self-trapped state induced absorption is completely suppressed and invisible in the absorption spectrum.**

Previous study has reported that the PL spectra of the (BA)₂PbI₄ exhibit three distinguish peaks below the free exciton at 4K (Fig. R2a, [Nature Communications. 9, 2254. (2018)]). The power dependent spectra show that all those three peaks exhibit a nearly linear dependence on the excitation power (Fig. R2b), indicating an intrinsic origin of these emission peaks rather than defects or bi-excitons [Nature Communications. 9, 2254. (2018)]. Thus, both peaks might be due to the self-trapped states but formed at different sites (*e.g.* Pb₂³⁺, Pb³⁺ and I²⁻) according to previous report [J. Mater. Chem. C, 5, 2771-2780. (2017)]. Similar spectrum also has been reported in organic material, *e.g.* perylene, from which rather narrow emission peaks due to the transition from the lowest self-trapped states to the intramolecular vibrational levels (Fig. R2c, [J. Phys. Soc. Jpn. 53, 3999. (1984)]).

[Redacted]

Fig. R2. (a) PL spectra of $(\text{BA})_2(\text{MA})_{n-1}\text{Pb}_n\text{I}_{3n+1}$ at 4K. Self-trapped states below free exciton are marked by dash lines. **(b)** Power dependent PL intensity of $(\text{BA})_2\text{PbI}_4$. Free exciton, low energy shoulder and self-trapped states are marked by red circles, black squares and blue triangles respectively. **(a)** and **(b)** are taken from [Nature Communications. 9, 2254. (2018)]. **(c)** Temperature dependent luminescence spectra of β -perylene. Free exciton is labeled by F and self-trapped states from different bands are labeled by ST(d) or ST(s). Footnote number indicate different intramolecular vibrational level (0, 1, 2...). **(c)** is taken from [J. Phys. Soc. Jpn. 53, 3999. (1984)].

Fig. R3a and 3b display the PL spectrum for $(\text{BA})_2\text{PbI}_4$ and $(\text{BA})_2(\text{MA})\text{Pb}_2\text{I}_7$ at 77 K, respectively. Based on above discussion, we attributed the emission peaks below free exciton more than 100 meV in $(\text{BA})_2\text{PbI}_4$ microplates (Fig. R3a) to the different type of self-trapped (ST) states, and labelled them as ST1, ST2 and ST3 according to the results in [Nature Communications. 9, 2254. (2018) and J. Am. Chem. Soc. 137, 2089–2096. (2015)]. In $(\text{BA})_2(\text{MA})\text{Pb}_2\text{I}_7$ microplates (Fig. R3b), we only observe two peaks below free exciton and the third one may be too weak to be observed at 77 K. In absorption spectrum, the ST1 associated absorption peak has also been observed while ST2 peak shows a rather weaker intensity than ST1 for both $(\text{BA})_2\text{PbI}_4$ and $(\text{BA})_2(\text{MA})\text{Pb}_2\text{I}_7$ microplates (Fig. R3c). Similar absorption spectrum also has been reported in thin film by previous work and assigned to the self-trapped state induced absorption as shown in Fig. R1e [ACS Nano. 11, 10834–10843. (2017)]. Remarkably, the absorption peak of ST1 can sustain up to room temperature and matches very well with the reflection peak of self-trapped states and the narrowband photoresponse peak (Fig. R3d, e), suggesting that they originate from the same optical transition: self-trapped states. Furthermore,

we also extracted the intensity ratio of self-trapped states associated reflection peak to the free exciton associated reflection peak and found that the ratio continuously decreases with the increase of the layer number n (Fig. R3f), which is another evidence for the presence of the self-trapped states since the trap states prefer to be localized at the interfaces of 2D perovskites [J. Am. Chem. Soc. 137, 2089–2096. (2015)].

Fig. R3. (a, b) PL spectrum of $(\text{BA})_2\text{PbI}_4$ (a) and $(\text{BA})_2(\text{MA})\text{Pb}_2\text{I}_7$ (b) at 77 K. (a) and (b) are plotted in \log scale to magnify weak ST peaks. ST1, ST2 and ST3 represent the different type of self-trapped states probably formed at different sites (c) Absorption spectra of $(\text{BA})_2\text{PbI}_4$ and $(\text{BA})_2(\text{MA})\text{Pb}_2\text{I}_7$ microplates with thickness of ~ 100 nm. ST1 is marked by dash line. (d, e) Absorption, reflectance and photocurrent spectra of $(\text{BA})_2\text{PbI}_4$ (d) and $(\text{BA})_2(\text{MA})\text{Pb}_2\text{I}_7$ (e). The photocurrent peak marked by dash line matches with ST1 in absorption and reflectance spectra. (f) Normalized intensity ratio of self-trapped state associated reflection peak to the free exciton associated reflection peak as a function of the layer number n .

Second, as suggested by reviewer #1, we have measured the temperature dependent absorption spectra of $(\text{BA})_2(\text{MA})\text{Pb}_2\text{I}_7$ sample and extracted the Urbach slope to confirm the existence of the self-trapped states in our samples. Due to the coexistence of the mixed phases at low temperature in $(\text{BA})_2\text{PbI}_4$ sample, we only focus on the $(\text{BA})_2(\text{MA})\text{Pb}_2\text{I}_7$ samples here. Fig. R4 shows the temperature dependent absorption spectra in semi-logarithmic scale together with the Urbach edge fittings. The Urbach tail can be expressed as $\alpha(E) = \alpha_0 \exp\left(-\sigma \frac{E_0 - E}{kT}\right)$, where α is the absorption coefficient, k is Boltzmann constant, T is the temperature, σ is the empirical slope coefficient, E_0 and α_0 are fitting parameters. Base on the Urbach slope parameter σ , a key parameter g (defined by $\sigma = s/g$, where s is direct exciton edges determined from the numerical simulations (1.5 for 3D and 1.24 for 2D)) is introduced to characterized the electron-phonon coupling strength and thus used to predict the existence of self-trapped states. If g exceeds a

typical value, *e. g.* 0.92 for 3D and 0.87 for 2D, self-trapped states are prone to exist. **By fitting the slope of the absorption edge, we obtain that $\sigma = 0.6825$, $g = 1.817$ (larger than 0.87) and $E_0 = 2.27$ eV for $(\text{BA})_2(\text{MA})\text{Pb}_2\text{I}_7$ samples** [K.S. Song, R.T. Williams, Self-Trapped Excitons, 1996, Springer]. Then, the Urbach energy E_u (defined by $E_u = kT/\sigma$) is calculated to be ~ 38 meV, which is close to previous reported value of ~ 31 meV in $(\text{BA})_2\text{PbI}_4$ [ACS Nano. 11, 10834–10843. (2017)]. This result also confirms that self-trapped states are present in our 2D perovskites.

Fig. R4. Temperature dependent absorption of $(\text{BA})_2(\text{MA})\text{Pb}_2\text{I}_7$ and Urbach edge fitting of different temperature are plot in dash lines.

Third, we demonstrated that the observed additional reflection peak and PL tails are not originated from impurity phases, edge states and permanent defects at least for $(\text{BA})_2\text{PbI}_4$ and $(\text{BA})_2(\text{MA})\text{Pb}_2\text{I}_7$, as questioned by reviewer #1. For the edge states, previous study [Science 355, 1288-1292. (2017); Nano Lett. 18, 5603–5609. (2018)] reveals that **the edge states are absent in $(\text{BA})_2\text{PbI}_4$ and $(\text{BA})_2(\text{MA})\text{Pb}_2\text{I}_7$ samples**, suggesting that the observed additional reflection peak and PL tails are not from the edge states. For those samples with large layer number n , **it is still unlikely that the edge states play a dominant role since all samples we used are large crystals in absence of a large number of edges**. To exclude the additional peak originated from permanent defect, we have carried out temperature- (Fig. R5a) and power-dependent PL studies of $(\text{BA})_2(\text{MA})\text{Pb}_2\text{I}_7$ exfoliated microplates at low temperature (Fig. R5b). From Fig. R5a, we observe two peaks below free exciton at low-temperature and ST1 shows a much stronger intensity than ST2. Both ST1 and ST2 gradually become broaden and weak with the increase of the temperature and finally disappear at room temperature due to the reduced radiative recombination rate. **The PL intensity of the additional emission peak shows a linear increase with the excitation power and the PL peaks show no shift with the excitation power** (Fig. R5c), suggesting the additional peak are not from permanent defects. Otherwise, the power different PL intensity should show different slope [Nature Communications. 9, 2254. (2018)]. In addition, for all samples, **the reflection spectra show similar spectral profile, indicates the additional peak would be unlikely from the permanent defects**. In terms of impurity phases, **there is no such problem for $(\text{BA})_2\text{PbI}_4$ and our XRD pattern, absorption spectrum and PL spectrum all indicates that our $(\text{BA})_2(\text{MA})\text{Pb}_2\text{I}_7$ samples are quite pure**. For $(\text{BA})_2(\text{MA})\text{Pb}_2\text{Br}_7$ samples, the impurity phase has been identified as $(\text{BA})_2\text{PbBr}_4$ phase, which would not introduce the below-gap states.

Nevertheless, for $n=3, 4$ and 5 samples, we admit the presence of the impurity phases would possibly lead to the below-gap states, as pointed out by the referee #1. To sum up, the self-trapped states lead to the additional reflection peak below the gap and PL tails for $n=1$ and $n=2$ samples and impurity phases would contribute to the below-gap states for $n > 2$ samples as well. Nevertheless, we believe that the **self-trapped states still play the dominant role** for the narrowband response in our devices since the impurity phases have rather **strong optical transition strength compared with the self-trapped states** and majority of the excitons generated within the impurity phases would **recombine under such weak applied electric field** in view of the **energy band alignment which favors the energy funneling** [J. Am. Chem. Soc. 137, 7843–7850. (2015)]. In contrast, the self-trapped states with a much longer lifetime could contribute to the photocurrent. Nevertheless, the presence of impurity phase would broaden the FWHMs of the narrowband response, as has been observed in our experiments. We have added more discussion and revised the manuscript accordingly.

Fig. R5. Temperature- and power-dependent PL spectra of $(\text{BA})_2(\text{MA})\text{Pb}_2\text{I}_7$. (a) Temperature-dependent PL map of $(\text{BA})_2(\text{MA})\text{Pb}_2\text{I}_7$ microplate. Sub-bandgap emission peaks are labeled by ST1 and ST2, respectively. (b) Normalized PL spectra under different incident laser power. (c) Normalized PL peak intensity of free exciton (565 nm) and ST1 exciton (614 nm) versus excitation power.

2. The role of self-trapped states in the narrowband photoresponse.

In this section, we will show that the narrowband response can indeed attribute to self-trapped states induced by strong electron-phonon interaction in our samples based on the following three evidences. First, if there is no contribution from the self-trapped states, there would be only two response peaks in Config. 3 devices. Rather, we observed three peaks with one response peak far below the free exciton peak (Fig. R6a). Provided that this X_t peak originates from the band tail absorption like 3D perovskite case, we should only observe the broadened free exciton peak rather than the distinctive X_t peak far below free exciton peak X_o . Second, without the contribution from the self-trapped states, the EQE of the narrowband response would continuously decrease with the increase of the thickness of the samples as 3D perovskite case (Fig. R6b, [Adv. Mater. 29, 1602639. (2017)]). Nevertheless, we observed that the EQE first increases and then decrease with the thickness of the samples under the same electric field (Fig.

R6c), which agrees well with the ST1 peak in the reflection spectrum and absorption spectrum (Fig. R6d). This cannot be explained by bare exciton diffusion without considering the self-trapped states. Third, the narrowband response peak coincides well with the ST1 peak in absorption spectrum and reflection spectrum, suggesting that ST1 would contribute to the narrowband response. Based on the above discussion, we believe that the self-trapped states play an indispensable role in the narrowband photodetections in our 2D perovskite devices.

Fig. R6. (a) Normalized EQE spectra of $(\text{BA})_2(\text{MA})\text{Pb}_2\text{I}_7$ device with Config. 3. Blue arrow: band-to-band (B); green arrow: band-edge exciton (X_0); red arrow: self-trapped states (X_1). **(b)** Normalized EQE under the same applied electrical field. The data points were taken from [Adv. Mater. 29, 1602639. (2017)]. **(c)** Thickness dependence of $(\text{BA})_2(\text{MA})\text{Pb}_2\text{I}_7$ peak position under the same applied electrical bias and normalized EQE under the same applied electrical field. The red and black dashed lines are just used guide the eye. **(d)** Absorption and reflectance of $(\text{BA})_2(\text{MA})\text{Pb}_2\text{I}_7$ at room temperature. The positions of ST are marked with dash lines.

3. Performance of narrowband detectors.

In our original manuscript, we evaluated the specific detectivity based on the shot noise, which underestimates the noise current, as the referee said. Therefore, we have re-measured a few new devices to calculate the specific detectivity by directly measured noise current and to estimate the linear dynamic range from the power dependent photoresponse. The I - V curve shows a slight nonlinearity probably due to the different contact of ITO and Au (Fig. R7a). Under the 620-nm light illumination, the current exhibits a significant increase in contrast to negligible response under a 550-nm excitation, suggesting the excellent wavelength-selective photoresponse of as-grown 2D perovskites. (Fig. R7a). The optical switch characteristic reveals the excellent stability and reversibility of our devices (Fig. R7b). The on-off ratio can reach $\sim 10^3$ excited by a 620-nm monochromatic light with a power of $20 \mu\text{W cm}^{-2}$, which is larger than the reported value in 3D perovskite based narrowband photodetectors (Fig. R7b, [Nature Photonics. 9, 679-686. (2015)]). The rise and falling time, defined as the time taken for the photocurrent increasing from 10 % to 90 % of the peak value and vice versa, were evaluated to be around 100 ms (Fig. 7c). We

also recorded the response current under different modulation frequency and the 3 dB cut-off frequency of the device is evaluated to be ~ 20 Hz (Fig. R7d). Interestingly, the spectral response exhibits a single narrow peak with the maximum EQE of $> 300\%$ even under a bias of 10 V, corresponding to an applied electric field of 0.12 V/ μm (Fig. 7e). With the increase of the bias, the EQE shows a gradual increase with the peak position maintaining the same. To evaluate the specific detectivity, we directly measured the noise of our detectors under a bias of 5 V at different frequency (Fig. R7f). The value of the noise current is ~ 0.06 pA Hz $^{-1/2}$, which is larger than the shot noise limit (~ 3.6 fA Hz $^{-1/2}$) calculated by the formula: $I_{s,n} = (2qI_d)^{1/2}$ (q is the value of electron charge and I_d is the dark current). Noise equivalent power (NEP) was calculated by the formula $\text{NEP} = P/(I_s/I_n) = I_n/(I_s/P) = I_n/R$ [Nature Photonics. 9, 687-694. (2015)], where P is incident light power intensity, R is the responsivity, I_s and I_n are photocurrent and noise current respectively. NEP is calculated to be ~ 0.1 pW Hz $^{-1/2}$ at the wavelength of 620 nm with a modulation frequency of 5 Hz. Afterwards, the detectivity D^* is calculated by $D^* = (AB)^{1/2}/\text{NEP}$ (A is the area of photodetector and B is the bandwidth). As shown in Fig. R7g. The maximum detectivity under a bias of 5 V is evaluated to be $\sim 1 \times 10^{11}$ Jones (illuminated by a 620 nm monochromatic light with an power density of 20 $\mu\text{W cm}^{-2}$), about 5 times larger than that of 3D perovskite based narrowband photodetectors [Nature Photonics. 9, 679-686. (2015)]. The linear dynamic range was measured to be 89 dB by $\text{LDR} = 20 \times \log(I_s/I_d)$ under a 633-nm laser illumination (Fig. R7h). We have replaced Fig. 2 in original manuscript by the re-measured ones.

Fig. R7. Device performance of an-80- μm -thick $(\text{BA})_2(\text{MA})\text{Pb}_2\text{I}_7$ narrowband photodetectors.

(a) Dark current and photocurrent of a detector under a 550-nm monochromatic illumination of 5 $\mu\text{W cm}^{-2}$ and a 620-nm monochromatic illumination of 20 $\mu\text{W cm}^{-2}$. **(b, c)** The optical switching characteristics **(b)** and temporal response **(c)** of the device illuminated by a 620 nm monochromatic light with a density of 20 $\mu\text{W cm}^{-2}$. **(d)** Normalized response as a function of signal frequency. The 3dB cut-off frequency is ~ 20 Hz. **(e)** EQE spectra of the device under different biases. **(f)** Measured total noise under a bias of 5 V and calculated shot noise limit (blue dash line). **(g)** Specific detectivity (D^*) spectrum under 5 V bias with a modulation frequency of 5

Hz. **(h)** Linear dynamic range of the detector under a bias of 5 V.

Response to Review 1.

The authors reported filterless narrowband photodetectors based on 2D Ruddlesden-Popper layered perovskites. They fabricated single crystals of 2D perovskites with controlled layer numbers and compositions. Based on these perovskite single crystals, they realized narrowband photodetection with broad spectral tunability. Compared to their 3D counterparts, 2D perovskites sustain narrowband response with reduced device thickness. Details are as follows.

Response: We thank the reviewer for carefully reading our manuscript. We also appreciate the suggestions given by the reviewer. We would like to take the opportunity to list the changes we have made in the revised manuscript according to reviewer's suggestions.

Specific comments:

1) *The authors attributed the narrowband photodetection to the self-trapped states. However, I cannot find any solid evidence on the presence of self-trapped excitons. The only evidence presented in this manuscript is redshift peaks in the reflection spectra. However, these additional peaks (Fig. 3a) can be caused by the domains with higher layer numbers, edge states and permanent defects. The PL tails (Fig. 2f) can also originate from these factors. Actually, according to the published results, I think the exciton self-trapping does not occur in BA 2D perovskites. The systems with self-trapped excitons usually feature broad, Stokes-shifted PL emission and relatively long carrier lifetime. The interactions between the exciton and deformable lattice will induce the loss of exciton energy and elongation of carrier lifetime. The single crystals of BA layered perovskites exhibit narrow PL emission, small Stokes shift and short lifetime of less than 10 ns (Science 355, 1288-1292 (2017)). However, I can accept the concept of self-trapped states if authors can perform more rigorous experiments. For example, the Urbach edge slope coefficient extracted by the temperature-dependent absorption can be employed to identify the exciton self-trapping (K.S. Song, R.T. Williams, Self-Trapped Excitons, 1996, Springer).*

Response: We thank a lot for the excellent points and suggestions. We have discussed all those relevant concerns and provide more evidences to prove that self-trapped states are indeed present in our samples at the beginning of this reply. We have added more discussion on this in the revised manuscript.

2) *Even without the concept of self-trapped states, the narrowband photodetection can be explained. The authors can consider the limited exciton diffusion length induced by the organic hopping barriers in 2D perovskites. With limited diffusion length, the excitons localized on the crystal surface, which generated by high-energy photons, are difficult to diffuse onto electrodes before recombination in config. 1 and 3 devices. To evaluate this mechanism, the authors can measure the exciton diffusion length in single crystals of 2D perovskites.*

Response: We thank the referee for this excellent point. Indeed, if we only explain a single EQE

spectrum of the device with a certain thickness in config.1 and config.2, the limited diffusion length model proposed by the referee can explain our results. Nevertheless, this model cannot explain the thickness dependent EQE in config.1 under the same electric field (See Fig. R3a). In addition, based on this model, the distinctive three peaks in thick devices with config. 3 cannot be interpreted. Only with introducing the self-trapped states, all those results can be coherently explained, which we have discussed in details at the beginning of this reply.

It is indeed helpful to verify this mechanism by extract the exciton diffusion length. Unfortunately, as we have neither the necessary facilities nor sophistication in ultrafast spectroscopy, we are not able to experimental extract the exciton diffusion length. Nevertheless, we have estimated the exciton diffusion length (L_D) from data in literature with the formula: $L_D=(D*\tau)^{1/2}$, where D is diffusion constant and τ is the lifetime [J. Phys. Chem. A. 109, 5271. (2005)]. The life time of $(BA)_2PbI_4$ is several hundred picoseconds [ACS Nano. 10, 9992–9998. (2016)]. The value of D can be estimated from the mobility which is $\sim 1 \text{ cm}^2 \text{ V}^{-1} \text{ s}^{-1}$ in $(BA)_2PbI_4$ [Nano Lett. 16, 7001–7007. (2016)]. Based on these values, the diffusion length of the exciton in $(BA)_2PbI_4$ is estimated to $\sim 100 \text{ nm}$. With the increase of the layer number n , the exciton diffusion length may slightly elongate due to the increment in both mobility and lifetime [Nano Lett. 16, 7001–7007. (2016); ACS Nano. 10, 9992–9998. (2016)]. Such small exciton diffusion length can indeed support the reviewer's suggestion that the narrowband response might be due to the limited diffusion length of excitons rather than self-trapped states. Nevertheless, limited diffusion length model cannot explain the thickness dependent EQE in config.1 under the same electric field and cannot explain three peaks in thick devices with config. 3. Therefore, we believe that the narrowband response in our devices should be attributed to the self-trapped states.

3) *The authors shows EQEs of exceeding 200 % in 2D-perovskite photodetectors. For a device with photodiode configuration, this represents a large photoconductive gain. The authors should carefully study and discuss the gain mechanism.*

Response: Thanks the reviewer for pointing this out. Indeed, if there is no applied bias, the EQE of the photodiode cannot surpass 100%, as the referee said. Nevertheless, the EQE can be very large if the device is forward biased, which has been observed in a number of reports [Adv. Mater. 27, 1912-1918. (2015); Adv. Mater. 28, 8144-8149. (2016)]. In fact, due to the large resistance in the out-of-plane direction, the contact barrier does not play a dominant role in the output characteristics of our devices. As you can see, even we used different contacts of ITO and Au, only slight asymmetry of IV curves has been observed. Under such case, our devices function more like photoconductors rather than photodiode. Thus, The EQEs of our devices can exceed 200% under a forward bias. We have clarified this in the revised manuscript.

4) *The authors show nearly symmetric IV curve in Fig. 2b. However, the device configuration is photodiode with different contacts of ITO and Au. The presence of Schottky barrier should lead to the asymmetry of IV curves. The authors need to carefully check their data or provide some explanation.*

Response: We thank the reviewer for pointing this out. As mentioned above, the extreme large resistance along the out-of-plane direction dominate over the contact resistance, leading to only a slight asymmetry of IV curves. As a matter of fact, the degree of asymmetry of IV curves depends

on the thickness of the devices. For the thinner devices, IV curves exhibit a large asymmetry (Fig. R8a and Fig. R8b). We have discussed this more in the revised manuscript.

Fig. R8. IV curve of an 80- μm -thick device **(a)** and a 5- μm -thick device **(b)** with or without illumination.

5) *The XRD peaks ranging from 2° to 15° are very important to identify the layer numbers in 2D perovskites. However, in Fig. 1c, these peaks is not presented clearly, especially for $n=4$, 5 perovskites. The $n=3$ perovskites contain some $n=4$ phase, which was not mentioned in manuscript. The authors need to pay more attention to the preparation of 2D perovskites with pure phase (Pb_2Br_7 and Pb_3I_{10}). The phase purity is important for the study of physical properties and the photodetectors with narrower spectral response.*

Response: We thank a lot the reviewer for carefully reading our manuscript and pointing this out. To make it clear, we replotted Fig. 1c and magnified the low angle part of XRD spectra (Fig. R9). As the referee said, for I-based $n>2$ perovskites, they indeed contain impurity phases, which would affect FWHMs of the narrowband photodetectors. The main peaks agree well to the diffraction pattern according to previous report [Science 355, 1288-1292 (2017)]. For Br-base $n=2$ perovskite, it also contains the impurity phase of $n=1$; nevertheless, it would introduce the in-gap states and thus would not affect the FWHM of the narrow spectral response. We agree with the referee that the phase purity is critical for the study of the physical properties and we have tried very hard to get pure phase perovskites with $n>2$ but failed. We have added more discussion on this in the revised manuscript.

Fig. R9. Powder XRD spectra of the perovskite crystal plates, the low angle part is magnified.

Black dot: $(\text{BA})_2\text{PbBr}_4$ phase within $(\text{BA})_2(\text{MA})\text{Pb}_2\text{Br}_7$; Navy rhombus: $(\text{BA})_2(\text{MA})_3\text{Pb}_4\text{I}_{13}$ phase within $(\text{BA})_2(\text{MA})_2\text{Pb}_3\text{I}_{10}$.

6) *The FWHM and peak position of spectral response of photodetectors are dependent on the thickness of crystals. However, I cannot find any information about the crystal thickness in Table 1. The authors are required to study the FWHM and peak positions on each 2D perovskites with different thicknesses, instead of providing values without sample information.*

Response: We thank the referee for pointing this out. We have added the thickness information in Table 1 in the revised manuscript.

7) *Fig. 4i and Fig. S6 provide thickness-dependent peak positions. The authors seemed to fit these data points by a linear function. What is the physical model used in these fitting?*

Response: Thank the reviewer for pointing this out. We did not use any physical model to fit those data points. The lines are only used to guide the eye. We have added this information in the figure caption in the revised manuscript.

8) *This manuscript provides a relatively high detectivity of 10^{13} Jones. The authors should provide the value of light power for calculating this detectivity. Additionally, power-dependent photocurrents and responsivities are required. The linear dynamic range of these photodetectors need to be calculated.*

Response: We very appreciate the suggestions by the referee. We have re-measured the devices with the method the referee mentioned to evaluate the detectivity. We also included the measurement of the power-dependent photocurrent and calculated linear dynamic range in the revised manuscript (see the general statement at the beginning of this reply).

9) *The detectivity was calculated by shot noise limit and steady-state photocurrents. Therefore, this value was overestimated. The authors need to rigorously calculate the specific detectivities by measuring the noise currents.*

Response: Thanks the reviewer for pointing this out. We have directly measured the noise current of the device and re-calculated the detectivity. Indeed, the measured noise is larger than the shot noise (see the general statement at the beginning of this reply). We have replaced the Fig. 2 with newly measured one.

In summary, I cannot recommend this manuscript for the publication of Nature Communication at the present stage. However, I think I can reconsider the revised manuscript on condition, if authors could provide rigorous demonstration on the self-trapped states and could added systematic measurements on the figures of merit (spectral response, power-dependent photocurrents and noises) of devices

Response: We thank the referee again for reading our manuscript and raising critical comments to improve our manuscript. With these point-to-point replies and the extended experiments (*i.e.* temperature-dependent absorption, power-dependent photocurrent, spectral response, linear dynamic range and noises measurements), we hope we have met all the specific requirements

asked by the honorable reviewers for consideration for publications.

Response to Review 2.

In their manuscript “Self-trapped states enabled filterless narrowband photodetections in 2D layered perovskite single crystals” Li et al report the filterless narrowband 2D perovskite single-crystalline photodetectors enabled by self-trapped states mechanism. The device performance is very promising compared to previous 3D perovskite narrowband photodetectors and brings new insights into applications of 2D perovskite multi-quantum well single crystals. This study may sprout future researches on 2D halide perovskite monocrySTALLINE photodetectors and benefit broad scientific community. Nevertheless, the “Self-trapped states” mechanism proposed by the author remains arguable. The authors are suggested to provide additional evidences and proper physical interpretation illustrating such a mechanism. Besides, there are a few issues the authors should also consider addressing.

Response: We thank the reviewer for considering our manuscript and recognizing that our work ‘*may sprout future researches on 2D halide perovskite monocrySTALLINE photodetectors and benefit broad scientific community*’. We have carried out some extended experiments to strength our conclusions in our revised manuscript according to reviewer’s suggestions (See the general statements at the beginning of this reply). We especially appreciate the detailed suggestions/questions raised by the reviewer, and have revised our manuscript accordingly.

Specific comments:

1) *Some sentences are in bold with no reason, e.g., line 88th-92nd in page 4.*

Response: We thank the referee for pointing this out. We have changed those sentences back to normal font.

2) *Page 6, line 119-120, “...which can be attributed to the reduced quantum confinement effect for larger n number...” is not strictly correct. The bandgap evolution is a result of many-body interaction and exciton binding energy change through both quantum and dielectric confinement upon layer number n increase. The author is suggested to give a complete statement.*

Response: Thanks for pointing this out. According to the previous reports, the dielectric confinement enhances the exciton binding energy in 2D perovskites resulting in a strong exciton absorption and emission peak [Phys. Rev. B. 45, 6961-6964. (1992)] while the redshift of the emission peaks of perovskites is a result of reduced quantum confinement together with the reduced dielectric confinement as the layer number n increases [J. Am. Chem. Soc. 137, 7843–7850. (2015)]. We have corrected and emphasized this in our revised manuscript.

3) *Figure 1b, all the samples exhibit blueshift of PL peak from their absorption edge. While typically the photon energy measured via PL is lightly smaller than the optical bandgap obtained from the absorption edge by an amount of the binding energy, that should be redshift. The author needs to provide some information or discussions based on this concern. Figures such as Stokes shift as a function of n might be helpful.*

Response: We thank the referee for the excellent point. **Since the crystals we used to measure the absorption spectra are very thick (around tens of micrometers), the absorption has already saturated before the free exciton absorption peak appears.** This is the reason we only can see a flat absorption line at short-wavelength regime. As a result, it seems that the PL peak shows a blueshift compared with that of absorption peak. **If we exfoliated microplates with thickness ~ 100 nm, we can observe the free exciton absorption peak and the PL peak shows slightly redshift (11 meV for $n = 2$ and 18 meV for $n = 1$) compared with the exciton absorption peak** (See Fig. R10). Nevertheless, the absorption peak relies on the thickness of the microplates, which makes it hard to compare the Stokes shift among different n . We have added this discussion in the revised manuscript.

Fig. R10. Absorption and PL spectra of $(\text{BA})_2\text{PbI}_4$ (a) and $(\text{BA})_2(\text{MA})\text{Pb}_2\text{I}_7$ (b) microplates with thickness of ~ 100 nm at room temperature.

4) Page 8, line 165-167, “...the external quantum efficiency in our devices is more than two orders of magnitude higher than that in 3D perovskite single crystal narrowband photodetectors...” The EQE can be magnified significantly through enlarging the driving voltage through multiple mechanisms. In the cited reference of 3D perovskite single crystal narrowband photodetectors (*Nature Photonics* 9, no. 10 (2015): 679-686), the driving voltage is 4 V and an electric field of ~ 0.01 V/ μm can induce a 20% EQE. Then the 400% EQE in this report (under a bias of 10 V, corresponding to an applied electric field of 0.12 V/ μm) seems not a big enhancement.

Response: We agree with the referee that ‘The EQE can be magnified significantly through enlarging the driving voltage’. From this aspect, the EQE of 400% is indeed not a big enhancement if we normalize this to the applied electric field. Nevertheless, as we have mentioned in the manuscript, **one big advantage of our devices is that we can applied a bigger electric field or we can make the device thinner (as thin as 5 μm) compared with that of 3D perovskite (as thick as 300 μm) based narrowband photodetectors** due to the extreme large resistance along the out-of-plane direct in 2D perovskites. Under such case, for 3D perovskite based narrowband photodetectors, the applied field cannot be too large or the device cannot too thin before they become broadband photodetectors (Fig. R11, [*Nature Photonics*. 9, 679-686. (2015)]). In our 2D perovskite based narrowband photodetectors, **the device can sustain two**

orders of magnitude higher electric field for devices with the same thickness as 3D perovskite devices, leading to a much larger EQE. We have illustrated this more in the revised manuscript.

[Redacted]

Fig. R11. (a) EQE spectra of a 0.3-mm-thick MAPbBr₃ single crystal under different reverse biases. **(b)** EQE spectra of MAPbBr₃ single crystals with different thicknesses under -4 V bias. The figures are taken from [Nature Photonics. 9, 679-686. (2015)].

5) *Page 9, In authors' assumption, the narrow photoresponse comes from the "low absorption coefficient of the sub-bandgap absorption" as it provides bulk-generated carriers that are hard to recombine. Then the sub-band states are majorly responsible for the photoresponse. Since single crystals are relatively pure with less sub-band states compared to polycrystals, as those single crystal devices perform quite well, how about the polycrystal devices? In addition, the self-trapped states in white LED the author cited in Page 4 have wide distributions so that the emitting light span a wide range of wavelength. Interestingly, the sub-band states assumption in this study delivers a narrow photoresponse. The author needs to give some clarifications to make the consistence with previous reports.*

Response: Thanks the reviewer for this excellent point. We did not make polycrystalline devices at the present work, but this is very interesting question and it is worth making such a comparison. We expect that the polycrystal devices can also achieve a narrowband photoresponse, which has been demonstrated in a 3D perovskite film device but at a sacrifice of FWHMs [Adv. Mater. 29, 1602639 (2017); Nature Photonics. 9, 687-694. (2015)]. For the 2D perovskite polycrystal devices, they might exhibit quite different behavior due to the presence of the organic cations and the presence of grain boundaries might be beneficial for the carrier extraction and thus increase the EQE of narrowband photodetectors but would also broad the FWHMs of devices. Nonetheless, further investigations are required to clarify this.

In terms of the self-trapped states in 2D perovskites, we have carried out a series of experiments to support the role of self-trapped states playing in our narrowband devices in the revised manuscript and discussed in details at the beginning of this reply. The self-trapped states in 2D perovskites strongly depend on the organic spacers, which leads to the different electron-phonon coupling strength and thus the degree of the lattice distortion. While strong broad band white-light emission has been observed for some sorts of 2D perovskites, self-trapped states

could give rise to multiple emission peaks with relative narrow bandwidth or weak emission peak in other sorts of 2D perovskites [J. Mater. Chem. C, 5, 2771-2780. (2017)]. In our $(\text{BA})_2(\text{MA})_{n-1}\text{Pb}_n\text{I}_{3n+1}$ series, two or three emission peaks (ST1, ST2 and ST3) originated from self-trapped states are observed while ST1 associated absorption peak shows a stronger absorption spectrum than ST2 and ST3. The ST2 and ST3 states only presented as a long tailing in the absorption spectrum due to the rather weak optical transition strength (Fig. R3). Nevertheless, the narrowband response of our devices is from the ST1 absorption, which has a relatively narrow bandwidth. Thus, we experimentally observed a narrowband response in our devices. We have added discussion on this in the revised manuscript and thank the referee again for this excellent point.

Fig. R3. (a, b) PL spectrum of $(\text{BA})_2\text{PbI}_4$ (a) and $(\text{BA})_2(\text{MA})\text{Pb}_2\text{I}_7$ (b) at 77 K. (a) and (b) are plotted in \log scale to magnify weak ST peaks. ST1, ST2 and ST3 represent the different type of self-trapped states probably formed at different sites (c) Absorption spectra of $(\text{BA})_2\text{PbI}_4$ and $(\text{BA})_2(\text{MA})\text{Pb}_2\text{I}_7$ microplates with thickness of ~ 100 nm. ST1 is marked by dash line. (d, e) Absorption, reflectance and photocurrent spectra of $(\text{BA})_2\text{PbI}_4$ (d) and $(\text{BA})_2(\text{MA})\text{Pb}_2\text{I}_7$ (e). The photocurrent peak marked by dash line matches with ST1 in absorption and reflectance spectra. (f) Normalized intensity ratio of self-trapped state associated reflection peak to the free exciton associated reflection peak as a function of the layer number n .

6) Page 9, line 196,197, "...When the wavelengths are far below the optical gap of the 2D perovskites, the absorption coefficient decreases further to nearly zero and thus the photoresponse can be neglected ..." it is confusing why absorption coefficient decreases to zero at far short wavelength region, as in the absorption spectrum, the absorbance keeps invariant in the region below optical bandgap.

Response: Thanks the referee for pointing this out. It is our mistake to describe ‘the long wavelength range’ as ‘wavelengths are far below the optical gap’. As can be seen in Fig. R6d, the absorbance decreases to zero at the long wavelength range far away from the free exciton absorption peak. We have corrected that in revised manuscript.

Fig. R6d Absorption and reflectance of $(\text{BA})_2(\text{MA})\text{Pb}_2\text{I}_7$ at room temperature. The positions of ST are marked with dash lines.

7) Page 10, line 218, typo “blow-gap”.

Response: Thanks for pointing his out. We have corrected this typo in our revised manuscript.

Response to Review 3.

This manuscript by Li et al. reported the filterless narrowband photodetector based on 2D layered perovskite single crystal. The authors did a comprehensive work on identifying the effect of the self-trapped states, which is unique to the 2D layered perovskite, on the device operation mechanism of the narrowband photodetectors. However, the basic working principle of the device is still the charge collection narrowing induced narrowband photon response which is not new. Moreover, I remain suspicious of whether the self-trapped states is indeed good for narrowband photodetection, considering the broad energetic distribution of these below gap states that will inevitably increase the Urbach energy and thus giving rise to a large FWHM of the photon response spectrum. In fact, the FWHM of the current narrowband photodetector based on 2D perovskite single crystal is still considerably larger than that of the photodetector made from 3D perovskite single crystal reported previously (see Ref. 12). In addition, the authors claimed orders of magnitude higher sensitivity of the 2D perovskite single crystal based photodetectors compared to the 3D perovskite based ones, which unfortunately is based on a wrong characterization method. They assumed the shot noise from dark current is the major contributor to the total noise, which however is not true in most cases. I suggest the authors directly measure the noise, and do the linear dynamic range characterization to verify the high sensitivity of their photodetectors. Therefore, in my opinion this work may be more suitable for a specialized journal after revisions, but lacks sufficient novelty and context for a high profile journal like Nature Communications.

Response: We thank the reviewer for carefully reading our manuscript. We appreciate the reviewer's comments and respectfully argue that our work shows a novelty and importance for the perovskite material community.

We agree with the referee that the mechanism we use in our narrowband is charge collection narrowing (CCN), which has been first developed in organic photodiodes in red and near-infrared wavelength range [Nature Communications. 6, 6343, 2015] and later was applied to narrowband visible photodetectors based on the mixture of 3D organohalide perovskites and organic molecule [Nature Photonics. 9, 679-686. (2015)]. Nevertheless, we should notice that the EQE measured in previous work is rather low due to the low absorption coefficient below band-gap and the response bandwidth is close to 100 nm. The mixed organic molecules with perovskites to serve as an extra absorption onset was used to enhance the EQE of narrowband photodetectors [Nature Photonics. 9, 687-694. (2015)], which however increases the complex of the device fabrication and sacrifices the bandwidth of the photoresponse. **It is worth noting that each one of these material systems has its own unique potential and also unique sets of fundamental challenges to overcome to achieve a breakthrough that can lead to technological innovation.** Our study represents the first attempt to fabricate narrowband photodetectors based on 2D perovskites, which is fundamentally important and practically relevant for the 2D perovskite community.

Furthermore, the self-trapped states in 2D perovskites are naturally formed and can enhance the below-gap absorption. With such an advantage, we can fabricate narrowband detectors with a

high EQE without complex material designing. Besides, due to the large resistance in the out-of-plane direction in 2D perovskites compared with 3D analogues [ACS Nano. 12, 4919-4929. (2018)], the narrowband response **maintains under a large electric field** and thus we can make the narrowband photodetectors to be **much thinner around 5 μm** (Figure S1), which is impossible in 3D perovskite based narrowband photodetectors. Therefore, we think our 2D perovskite-based narrowband detectors show some outstanding merits that are missing in 3D perovskite-based devices and thus our work could make a significant contribution to the 2D perovskite material community as claimed by referee #2. **Although previous reports can provide some insights for our work, we believe that they don't affect the quality and significance of our study and the novelty of our central conclusions and our study has still its own high quality and significance in the field of 2D perovskite studies.**

In terms of whether the self-trapped states are good for narrowband detection, we have carried out a series of experiments to support the role of self-trapped states playing in our devices in the revised manuscript and discussed in details above. The self-trapped states strongly depend on the organic spacers, which leads to the different electron-phonon coupling strength and thus the degree of the lattice distortion. While strong broad band white-light emission has been observed for some sorts of 2D perovskites, the self-trapped states could give rise to multiple emission peaks with relative narrow bandwidth or weak emission peak in other sorts of 2D perovskites. In our $(\text{BA})_2(\text{MA})_{n-1}\text{Pb}_n\text{I}_{3n+1}$ series, two or three emission peaks (ST1, ST2 and ST3) originated from self-trapped states are observed while ST1 associated absorption peak shows a stronger absorption spectrum than ST2 and ST3. The ST2 and ST3 states only presented as a long tailing in the absorption spectrum due to the rather weak optical transition strength (Figure R3). Nevertheless, **the narrowband response of our devices is from the ST1 absorption, which has a relatively narrow bandwidth.** Thus, we observed a narrowband response in our devices.

Compared with 3D perovskite single crystal based narrowband photodetectors [Nature Photonics. 9, 679-686. (2015)], the spectral response bandwidth is **comparable or even narrower for our devices in the blue wavelength** range and indeed larger in the green and red wavelength range but **the response peak can extend to 690 nm.** Nevertheless, if we change the FWHMs from nm to meV, they have more or less the same bandwidth. In addition, the largest bandwidth in our 2D perovskite based narrowband photodetectors is around 60 nm for I-based $n=5$ devices, which is **still much narrower than that of reported narrowband photodetectors based on organic photodiode** [Nature Communications. 6, 6343, 2015] **and the mixture of organohalide perovskites and organic molecules** [Nature Photonics. 9, 687-694. (2015)].

As for the concern about the detectivity measurement, we have re-estimated it by directly measuring the noise current and discussed in details at the beginning of this reply.

Fig. R3. (a, b) PL spectrum of $(\text{BA})_2\text{PbI}_4$ (a) and $(\text{BA})_2(\text{MA})\text{Pb}_2\text{I}_7$ (b) at 77 K. (a) and (b) are plotted in \log scale to magnify weak ST peaks. ST1, ST2 and ST3 represent the different type of self-trapped states probably formed at different sites (c) Absorption spectra of $(\text{BA})_2\text{PbI}_4$ and $(\text{BA})_2(\text{MA})\text{Pb}_2\text{I}_7$ microplates with thickness of ~ 100 nm. ST1 is marked by dash line. (d, e) Absorption, reflectance and photocurrent spectra of $(\text{BA})_2\text{PbI}_4$ (d) and $(\text{BA})_2(\text{MA})\text{Pb}_2\text{I}_7$ (e). The photocurrent peak marked by dash line matches with ST1 in absorption and reflectance spectra. (f) Normalized intensity ratio of self-trapped state associated reflection peak to the free exciton associated reflection peak as a function of the layer number n .

Some typos:

Page 10, “blow-gap” should be “below-gap”;

Page 16, “CNN concept” should be “CCN concept”.

Response: Thanks the reviewer for pointing this out. We have corrected these typos in our revised manuscript.

Reviewers' comments:

Reviewer #1 (Remarks to the Author):

In this revised version, I think the authors have successfully improved the quality of the manuscript. They have added many experiments to demonstrate the self-trapped states and their roles in narrowband photodetection. The noise and frequency modulation measurements have also performed to enhance the rigidity of the device performance. I think this manuscript is now qualified for the publication of Nature Communications after some minor revisions.

i) The noise currents are not stable at low-frequency region. The authors are suggested to perform multiple measurements to calculate the mean values and standard deviation. The detectivities should be calculated based on the average noise current at same modulation frequency.

ii) The noise and frequency modulation measurements were carried out by lock-in amplifier and preamplifier? The authors need to provide more experimental details.

Reviewer #2 (Remarks to the Author):

The new experiments clearly add to the contribution of this paper, and most of my concerns from my previous review have been addressed. Minor corrections and clarifications are needed, as elaborated below.

1. Page 4, line 78, "self-trapped states below bandgap" should be "self-trapped states within bandgap" or "self-trapped states with below bandgap energy" and similar mistakes elsewhere.
2. Page 6, line 134, "it would introduce the sub-bandgap states". I guess the author wants to say "would not introduce states within the forbidden band".
3. To me, the sharp EQE response is a result of a combination of "bulk carrier recombination loss effect" and the so-called "self-trapped states effect". The author sells the "self-trapped states effect" quite well, but it is also suggested to give more credits to the "bulk carrier recombination loss" that leads to the shortwavelength EQE bleaching. Otherwise, the statement would be misleading to the broader audience.

Reviewer #3 (Remarks to the Author):

I appreciate the authors' efforts to address the comments. In the revised manuscript, they have followed the reviewer' suggestion to do the noise characterization and linear dynamic range measurement of the photodetector which gave rise to more accurate and convincing D^* value. However, the unique role played by the "self-trapped states" in narrowband photodetection and its merit are still not well explained. I'm convinced that the narrowband photon response of the 2D layered perovskite single crystal photodetectors are caused by the "self-trapped states" induced below gap absorption. However, the regular band-tail states in perovskites can also induce the below gap absorption that can be utilized to achieve the narrowband detection, and the corresponding Urbach tail is generally narrower in comparison to that with the "self-trapped states" (although the authors argue that for certain sorts of 2D layered perovskites, the ST1 absorption peak is also narrow). As a result, the uniqueness of the "self-trapped states" in realizing high performance narrowband photodetection and its advantage is not apparent. In fact, based on the model proposed by the authors, it seems that large resistance in the out-of-plane direction in 2D layered perovskites instead of the "self-trapped

states" is the key to achieve the lower dark current and higher tolerance to large electric field in 2D perovskite based narrowband photodetectors compared to those of the 3D perovskite single crystal based ones. I assume the narrowband photon response could still be maintained even if the "self-trapped states" were eliminated in 2D layered perovskites. One evidence is that the impact of the "self-trapped states" will be weakened with the increase of the n number in $(\text{BA})_2(\text{MA})_{n-1}\text{PbI}_{3n+1}$ single crystals, while the narrowband photon response performance is similar for the corresponding photodetectors. Therefore, in my opinion this work can be reconsidered for publication in Nature Communications only if the above comment can be appropriately addressed by the authors.

Minor issues:

1. The authors should do a fair comparison when referring to the device performance reported in the literature. For instance, the dark current density instead of the dark current should be compared since the latter is dependent on the device area; and the rejection ratio should also be compared which is an essential parameter for a narrowband photodetector.
2. It is surprising that the bulk generated excitons in 2D perovskites can be efficiently separated despite of their large exciton binding energy. The authors should at least provide some possible explanations considering it is critical to the understanding of the device operation mechanism.

Response to Reviewer #1

In this revised version, I think the authors have successfully improved the quality of the manuscript. They have added many experiments to demonstrate the self-trapped states and their roles in narrowband photodetection. The noise and frequency modulation measurements have also performed to enhance the rigidity of the device performance. I think this manuscript is now qualified for the publication of Nature Communications after some minor revisions.

Response: We thank the reviewer for feedback and helpful suggestions to improve the quality of our manuscript and for the support of publication in *Nature Communications*. We have further strengthened our manuscript according to the reviewer's suggestions.

Specific comments:

1) *The noise currents are not stable at low-frequency region. The authors are suggested to perform multiple measurements to calculate the mean values and standard deviation. The detectivities should be calculated based on the average noise current at same modulation frequency.*

Response: Thank the reviewer for the suggestion. We agree with the reviewer that 'the noise current is not stable at low-frequency'. In the last version of manuscript, we have already calculated the average value of noise current from ten-time measurements and displayed in Fig. 2f. The detectivities in Fig. 2f was actually already calculated based on the average noise current in the last version. In Fig. RR1, standard deviations were also calculated and added as error bars according to the reviewer's suggestion (Fig. RR1). We have emphasized on that and replaced the noise current figure with Fig. RR1 in our revised manuscript.

Fig. RR1. Average noise current measured under a bias of 5 V and calculated shot noise limit (blue dash line). Error bars are the standard deviations at each measurement frequency.

2) *The noise and frequency modulation measurements were carried out by lock-in amplifier and preamplifier? The authors need to provide more experimental details.*

Response: We thank the reviewer for pointing this out. As the reviewer said, both the noise and frequency modulation measurements are carried out by coupling a low-noise preamplifier with a lock-in amplifier. All filters of preamplifier have been switched off for the noise measurement to avoid the influence from those filters. We have added this important experimental details in our revised manuscript.

Response to Reviewer #2

The new experiments clearly add to the contribution of this paper, and most of my concerns from my previous review have been addressed. Minor corrections and clarifications are needed, as elaborated below.

Response: We thank the reviewer for these suggestions and have revised our manuscript accordingly.

Specific comments:

1) *Page 4, line 78, “self-trapped states below bandgap” should be “self-trapped states within bandgap” or “self-trapped states with below bandgap energy” and similar mistakes elsewhere.*

Response: We thank the reviewer very much for pointing this out. To be consistent, we have used ‘self-trapped states within bandgap’ to replace all ‘self-trapped states below bandgap’ throughout the manuscript in our revision.

2) *Page 6, line 134, “it would introduce the sub-bandgap states”. I guess the author wants to say “would not introduce states within the forbidden band”.*

Response: Thanks the reviewer for pointing this out. We have corrected this mistake in our revised manuscript.

3) *To me, the sharp EQE response is a result of a combination of “bulk carrier recombination loss effect” and the so-called “self-trapped states effect”. The author sells the “self-trapped states effect” quite well, but it is also suggested to give more credits to the “bulk carrier recombination loss” that leads to the short wavelength EQE bleaching. Otherwise, the statement would be misleading to the broader audience.*

Response: We thank the reviewer very much for this excellent point. I guess what the reviewer wants to say is ‘surface generated carrier recombination loss’, which leads to the short wavelength EQE bleaching as we have discussed in the manuscript. For the short wavelength range, the majority of charge carriers are generated in a narrow region near the surface (termed as surface generation) due to the large absorption coefficient induced small penetration depth according to the Beer-Lambert law. The collection of surface-generated carriers is suppressed via recombination losses due to the possible factors including the higher local carrier concentration, imbalanced transit times for the electrons and holes and the severe surface-charge recombination. As a result, the EQE at short wavelength range is bleached, leading to the sharp EQE response. We have revised the related expressions to make this much clear in the revised manuscript.

Response to Reviewer #3

I appreciate the authors' efforts to address the comments. In the revised manuscript, they have followed the reviewer's suggestion to do the noise characterization and linear dynamic range measurement of the photodetector which gave rise to more accurate and convincing D^ value. However, the unique role played by the "self-trapped states" in narrowband photodetection and its merit are still not well explained. I'm convinced that the narrowband photon response of the 2D layered perovskite single crystal photodetectors are caused by the "self-trapped states" induced below gap absorption. However, the regular band-tail states in perovskites can also induce the below gap absorption that can be utilized to achieve the narrowband detection, and the corresponding Urbach tail is generally narrower in comparison to that with the "self-trapped states" (although the authors argue that for certain sorts of 2D layered perovskites, the ST1 absorption peak is also narrow). As a result, the uniqueness of the "self-trapped states" in realizing high performance narrowband photodetection and its advantage is not apparent. In fact, based on the model proposed by the authors, it seems that large resistance in the out-of-plane direction in 2D layered perovskites instead of the "self-trapped states" is the key to achieve the lower dark current and higher tolerance to large electric field in 2D perovskite based narrowband photodetectors compared to those of the 3D perovskite single crystal based ones. I assume the narrowband photon response could still be maintained even if the "self-trapped states" were eliminated in 2D layered perovskites. One evidence is that the impact of the "self-trapped states" will be weakened with the increase of the n number in $(BA)_2(MA)_n-1PbnI_{3n+1}$ single crystals, while the narrowband photon response performance is similar for the corresponding photodetectors. Therefore, in my opinion this work can be reconsidered for publication in Nature Communications only if the above comment can be appropriately addressed by the authors.*

Response: We sincerely thank the reviewer for carefully reading our revised manuscript and giving some insightful comments. We also appreciated that the reviewer was convinced that 'the narrowband photon response of the 2D layered perovskite single crystal photodetectors are caused by the "self-trapped states" induced below gap absorption'. In the following, we will explain why the self-trapped states have to be there in order to explain our observed narrowband spectral response in 2D perovskites.

We agree with the reviewer that the narrowband response can be realized even if the self-trapped states were eliminated. Nevertheless, the presence of three distinctive photocurrent peak, thickness dependent EQE trend and enhanced EQE in our 2D perovskite devices cannot be well explained without the self-trapped states. First of all, if the regular band-tail states are considered to be the main origin of the narrowband response as suggested by the reviewer, we should only observe a **broadened free exciton peak rather than a distinctive X_1 peak more than 120 meV below X_0 peak** in Config. 3 (Fig. RR2a). Second, without the contribution from the self-trapped states, the EQE of the narrowband response would continuously decrease with the increase of the thickness of the samples as 3D perovskite case (Fig. RR2b). Rather, **we observed that the EQE in our 2D perovskite devices first increases and then decreases with thickness** (Fig. RR2c), which agrees well with the ST1 peak in the reflection spectrum and absorption spectrum. This also suggests that self-trapped states should be responsible to the narrowband response. Third, **we expect that the EQE of 2D perovskite devices would be much smaller than that of 3D counterparts under the assumption that regular band-tail states lead to the narrowband photoresponse, since the large resistance in the out-of-plane direction makes it more difficult for carriers to be**

extracted in 2D perovskites. In contrast, we observe an enhancement of EQE. Therefore, we believe that this enhancement should result from the enhanced absorption due to the presence of self-trapped states. In the revised manuscript, we illustrated more on why the self-trapped states have to take into account in order to explain our experimental results according to the above discussions.

Indeed, as the reviewer said, the impact of the self-trapped states will be weakened with the increase of the n number in $(\text{BA})_2(\text{MA})_{n-1}\text{Pb}_n\text{I}_{3n+1}$ crystals as seen from the reflection spectra. Thus, we should expect to observe a smaller EQE. Nevertheless, it should be noted that the photoconductivity process relates to **both the photocarrier generation and extraction**. With the increase of n , **while the intensity of self-trapped state assisted absorption is indeed weakened, the conductivity in the out-of-plane direction nevertheless continuously increases (Fig. RR2d taken from [ACS Nano. 12, 4919–4929 (2018)]), which would be beneficial for the carrier extraction**. Those two factors together result in the similar photon response performance in all $(\text{BA})_2(\text{MA})_{n-1}\text{Pb}_n\text{I}_{3n+1}$ crystals with different n . We have added more discussion on this in our revised manuscript.

[Redacted]

Fig. RR2. (a) Normalized EQE spectra of $(\text{BA})_2(\text{MA})\text{Pb}_2\text{I}_7$ device with Config. 3. Blue arrow: band-to-band (B); green arrow: band-edge exciton (X_0); red arrow: self-trapped states (X_t). (b) Normalized EQE of 3D perovskites under the same applied electrical field. The data points were taken from [Adv. Mater. 29, 1602639. (2017)]. (c) Thickness dependence normalized EQE of $(\text{BA})_2(\text{MA})\text{Pb}_2\text{I}_7$ under the same applied electrical field. (d) Electric conductance of 2D perovskites ($(\text{BA})_2(\text{MA})_{n-1}\text{Pb}_n\text{I}_{3n+1}$) and its 3D analogue (MAPbI_3). The figure is taken from [ACS Nano. 12, 4919–4929 (2018)].

. [Redacted]

[Redacted]

Specific comments:

- 1) *The authors should do a fair comparison when referring to the device performance reported in the literature. For instance, the dark current density instead of the dark current should be compared since the latter is dependent on the device area; and the rejection ratio should also be compared which is an essential parameter for a narrowband photodetector.*

Response: We thank the reviewer for those suggestions. The dark current density in our devices is estimated to be around $\sim 10^{-8}$ A cm⁻² with a thickness around 80 μ m. Based on the geometric size of their devices, the dark current density is estimated to be $\sim 10^{-7}$ A cm⁻² with a thickness of 1.2 mm in [Nature Photonics. 9, 679-686. (2015)]. Nevertheless, we think it would be not fair to compare the current density of the devices with different thickness and different device structure since the resistance of the device is related to thickness and a thick sample should result in a low current density. Even without considering the influence of thickness, the current density of 3D perovskites is still one order of magnitude larger than that in our devices. In terms of the rejection ratio, it is calculated to be ~ 40 in our devices, which is several times smaller than the reported values [Nature Photonics. 9, 679-686. (2015)]. We have added this discussion in the revised manuscript.

- 2) *It is surprising that the bulk generated excitons in 2D perovskites can be efficiently separated despite of their large exciton binding energy. The authors should at least provide some possible explanations considering it is critical to the understanding of the device operation mechanism.*

Response: The underlying mechanism of the bulk exciton ionization is not clear yet. As we have mentioned in the manuscript, the bulk generated excitons are inside the crystal, thus surface depletion field is not likely to be responsible for the self-trapped exciton ionization, unlike the free exciton case. One possibility is that the bulk excitons are separated at the interfaces between the organic layers and inorganic layers and then collected by electrodes under external applied electric

field, leading to the photocurrent. Nevertheless, further investigations are required to clarify this ionization mechanism. We have added this possibility in the revised manuscript.

REVIEWERS' COMMENTS:

Reviewer #3 (Remarks to the Author):

In the revised manuscript, the authors have added more discussion on the unique role of self-trapped states in realizing narrowband photon response in 2D perovskites, and elucidated clearly their difference with regular band-tail states. Therefore, my major concern about this work is well addressed by the authors, and now I can support the publication of this manuscript in Nature Communications.

Response to Reviewer 3

In the revised manuscript, the authors have added more discussion on the unique role of self-trapped states in realizing narrowband photon response in 2D perovskites, and elucidated clearly their difference with regular band-tail states. Therefore, my major concern about this work is well addressed by the authors, and now I can support the publication of this manuscript in Nature Communications.

Response: We sincerely thank the reviewer for the feedback and for the support of publication in *Nature Communications*.